https://doi.org/10.1038/s41467-021-24086-9　　**OPEN**

# Live-cell imaging of circadian clock protein dynamics in CRISPR-generated knock-in cells

Christian H. Gabriel [1,2], Marta del Olmo [3], Amin Zehtabian [4], Marten Jäger[5], Silke Reischl [1,2], Hannah van Dijk [1,2], Carolin Ulbricht [6,7], Asylkhan Rakhymzhan [8], Thomas Korte[9], Barbara Koller[1,2], Astrid Grudziecki[1,2], Bert Maier[1,2], Andreas Herrmann [9], Raluca Niesner [8,10], Tomasz Zemojtel[5], Helge Ewers [4], Adrián E. Granada [11], Hanspeter Herzel[3] & Achim Kramer [1,2✉]

The cell biology of circadian clocks is still in its infancy. Here, we describe an efficient strategy for generating knock-in reporter cell lines using CRISPR technology that is particularly useful for genes expressed transiently or at low levels, such as those coding for circadian clock proteins. We generated single and double knock-in cells with endogenously expressed PER2 and CRY1 fused to fluorescent proteins allowing us to simultaneously monitor the dynamics of CRY1 and PER2 proteins in live single cells. Both proteins are highly rhythmic in the nucleus of human cells with PER2 showing a much higher amplitude than CRY1. Surprisingly, CRY1 protein is nuclear at all circadian times indicating the absence of circadian gating of nuclear import. Furthermore, in the nucleus of individual cells CRY1 abundance rhythms are phase-delayed (~5 hours), and CRY1 levels are much higher (>5 times) compared to PER2 questioning the current model of the circadian oscillator.

[1] Charité Universitätsmedizin Berlin, corporate member of Freie Universität Berlin, Humboldt-Universität zu Berlin, and Berlin Institute of Health, Laboratory of Chronobiology, Berlin, Germany. [2] Berlin Institute of Health (BIH), Berlin, Germany. [3] Institute for Theoretical Biology, Charité Universitätsmedizin Berlin, Berlin, Germany. [4] Institute for Chemistry and Biochemistry, Department of Biology, Chemistry and Pharmacy, Freie Universität Berlin, Berlin, Germany. [5] Berlin Institute of Health (BIH) Core Genomics Facility, Charité Universitätsmedizin Berlin, Berlin, Germany. [6] Immune Dynamics, Rheumatology and Clinical Immunology, Charité – Universitätsmedizin Berlin, Berlin, Germany. [7] Immune Dynamics, Deutsches Rheuma-Forschungszentrum (DRFZ), a Leibniz Institute, Berlin, Germany. [8] Biophysical Analytics, Deutsches Rheuma-Forschungszentrum (DRFZ), a Leibniz Institute, Berlin, Germany. [9] Molecular Biophysics, Department of Biology, Humboldt Universität zu Berlin, Berlin, Germany. [10] Dynamic and Functional in vivo Imaging, Veterinary Medicine, Freie Universität Berlin, Berlin, Germany. [11] Charité Comprehensive Cancer Center, Charité Universitätsmedizin Berlin, Berlin, Germany. ✉email: achim.kramer@charite.de

To stay in synchrony with environmental cycles, most living organisms developed endogenous clocks, which regulate the circadian (~24 h) rhythms of molecular, physiological, and behavioral functions. The molecular basis of these circadian clocks is a gene-regulatory network with transcription-translation feedback loops driving cell-autonomous oscillations in most mammalian tissues. According to the current model, the heterodimeric transcription factor CLOCK:BMAL1 mediates the expression of rhythmically transcribed genes by binding to E-box enhancer elements in their promoters[1]. Among those genes are the canonical repressors PER1-3 and CRY1-2, which inhibit CLOCK:BMAL1 transcriptional activity after a delay of several hours and, thereby, their own expression[2,3]. After the regulated degradation of PERs and CRYs, the transcription factor complex is released from repression, and a new cycle can start[4–6]. Similar to BMAL1-knock-out animals, PER- or CRY-deficient mice are behaviorally arrhythmic, emphasizing the importance of each protein family for the integrity of the molecular clock[7,8].

PER and CRY proteins physically interact[9–11], and there is evidence that this interaction controls their subcellular localization. Genetic studies showed that CRYs do not accumulate in the nucleus of Per1/2 double knock-out cells and, similarly, PERs are almost exclusively cytoplasmic in cells lacking both CRY proteins[12] suggesting that the presence of each family is necessary for proper PER and CRY protein localization. This is supported by overexpression studies, in which CRY1 accelerates PER2 nuclear import dynamics in human cells[13]. In Drosophila, analogs of PERs and CRY, dPER and TIM, first accumulate in the cytoplasm when overexpressed and after a delay of several hours translocate into the nucleus together[14]. Although similar models were proposed for the mammalian system[15], no circadian differences in subcellular localization of PER2 were observed in cells from Per2-Venus knock-in mice[16], thus questioning this analogy.

Recent findings by Aryal et al.[17] indicate that PER proteins and most of CRY protein almost exclusively co-exist in huge cytosolic and nuclear complexes suggesting common regulation. Again, double knock-out of either Per1/2 or Cry1/2 completely prevented the formation of these complexes. Notably, however, minor but substantial amounts of the monomeric form of CRY1, but not of the other repressors were detected, in particular in the late repressive phase[17]. In addition, there is accumulating evidence that CRY1 plays a special role among the repressive proteins. Compared to PER1, PER2, and CRY2, whose expression appears to be synchronized, CRY1 mRNA and protein expression were reported to peak a few hours later in the circadian cycle,[18–20] and this delayed expression was found to be important for CRY1 to rescue rhythmicity in CRY1/CRY2 double knock-out cells[18,21–25]. Moreover, data from proteomic experiments also indicate that accumulation of CRY1 protein in the nucleus is delayed compared to the other repressors[26]. During the late repressive phase, CRY1 co-occupies BMAL1/CLOCK binding sites in the absence of CRY2 and PER proteins, demonstrating a PER-independent role of CRY1 in the nucleus[27]. From this, the concept of a functional distinct late repressive complex emerged, that contains only BMAL1/CLOCK and CRY1 and represents a DNA-bound, inactive but poised state[12,28–30].

Most of the current knowledge of PER and CRY protein dynamics resulted either from biochemical data with mixed lysates of many thousand cells, or from single-cell imaging of overexpressed fluorescent tagged fusion proteins[12,13,17,31]. Both approaches have clear limitations: population sampling – e.g. cell fractionation followed by western blot, chromatography, or immunoprecipitation – not only conceals spatial information, but also suffers from much reduced temporal resolution. Most importantly, however, population sampling averages signals from thousands of cells thereby masking individual cell properties (e.g.

regarding circadian period, phase, and amplitude) and degree of noise. While fluorescent tagged proteins constitute an outstanding tool to monitor protein expression and localization in individual living cells, overexpression of PER-and CRY-proteins in most cases disrupts the circadian oscillator and data from such experiments have to be interpreted with caution[32,33].

Such limitations can be overcome by incorporating a fluorescent tag directly into the proteins' genomic locus. In this case, expression dynamics and level of the resulting fusion protein often remain similar to the wild-type protein and the clock stays intact. Indeed, the Per2-Luciferase and the Per2-Venus knock-in mice – in which PER2 is tagged at the genomic level with a luciferase or a yellow fluorescent protein, respectively – enabled analysis of PER2 protein oscillations on a single-cell level without compromising the oscillator[16,34]. In contrast, expression and localization dynamics of endogenously expressed CRY proteins in live cells have not been reported yet, due to the lack of similar knock-in models. Furthermore, differences between the murine and primate circadian oscillator create a need for human cellular models[20]. Thus, this lack of model motivated us to create human cell lines that express fluorescence tagged versions of PER and CRY proteins from the respective endogenous loci.

While targeted introduction of DNA into the genome of a somatic cell used to be extremely inefficient and – if possible at all – laborious, the discovery and development of CRISPR/Cas9 based genome editing changed the game[35,36]. In short, targeted Cas9-mediated DNA double strand breaks are – among other possible outcomes – eliminated by the endogenous homology directed repair (HDR) pathway, which can be hijacked to introduce an exogenous donor sequence (such as a fluorescent protein tag) into the locus (reviewed in Singh et al.[37]). Although Cas9-induced double strand breaks greatly stimulate the integration of such a homologous donor, the rate of targeted integration is usually still low and depends on many parameters, such as cell type, transfection efficiency, and length of the integrated sequence. In addition, existing strategies to enrich for the desired cells are prone to fail when targeting genes that are expressed transiently or at low copy numbers.

Here, we report an efficient strategy to knock-in fluorescent reporter proteins into the genomic locus of the low copy number circadian proteins PER2 and CRY1 and applied it to human cells. We generated single and double knock-in cells with intact circadian clocks, which allowed us to monitor the dynamics of CRY1 and PER2 fusion proteins in live single cells. We found that CRY1 protein is mainly nuclear at all circadian times suggesting absence of circadian gating of nuclear import. Furthermore, CRY1 expression is phase-delayed and rises to much higher (>5 times) levels compared to PER2 protein in the nucleus of individual cells questioning the current model of the circadian oscillator.

## Results

**Strategy for CRISPR/Cas9-mediated knock-in.** To insert reporter protein tags into the PER2 and CRY1 genomic loci of human cells, we conceived a knock-in strategy for low copy number genes. Thereby various tags including mClover3[38] and mScarlet-I[39], bright monomeric green or red fluorescence proteins (FP), respectively, as well as firefly luciferase were aimed to be integrated into these loci. As an example, we outline the vector design as well as the screening strategy for generating knock-in cells that express PER2 C-terminally tagged to mScarlet-I from the endogenous PER2 locus in the following.

To this end, we intended to integrate the FP-coding sequence directly upstream of the STOP codon of PER2 (Fig. 1a). Hence, we designed single-guide RNAs (sgRNAs) that target the Cas9 to introduce double strand breaks very close (<60 bp) to the STOP

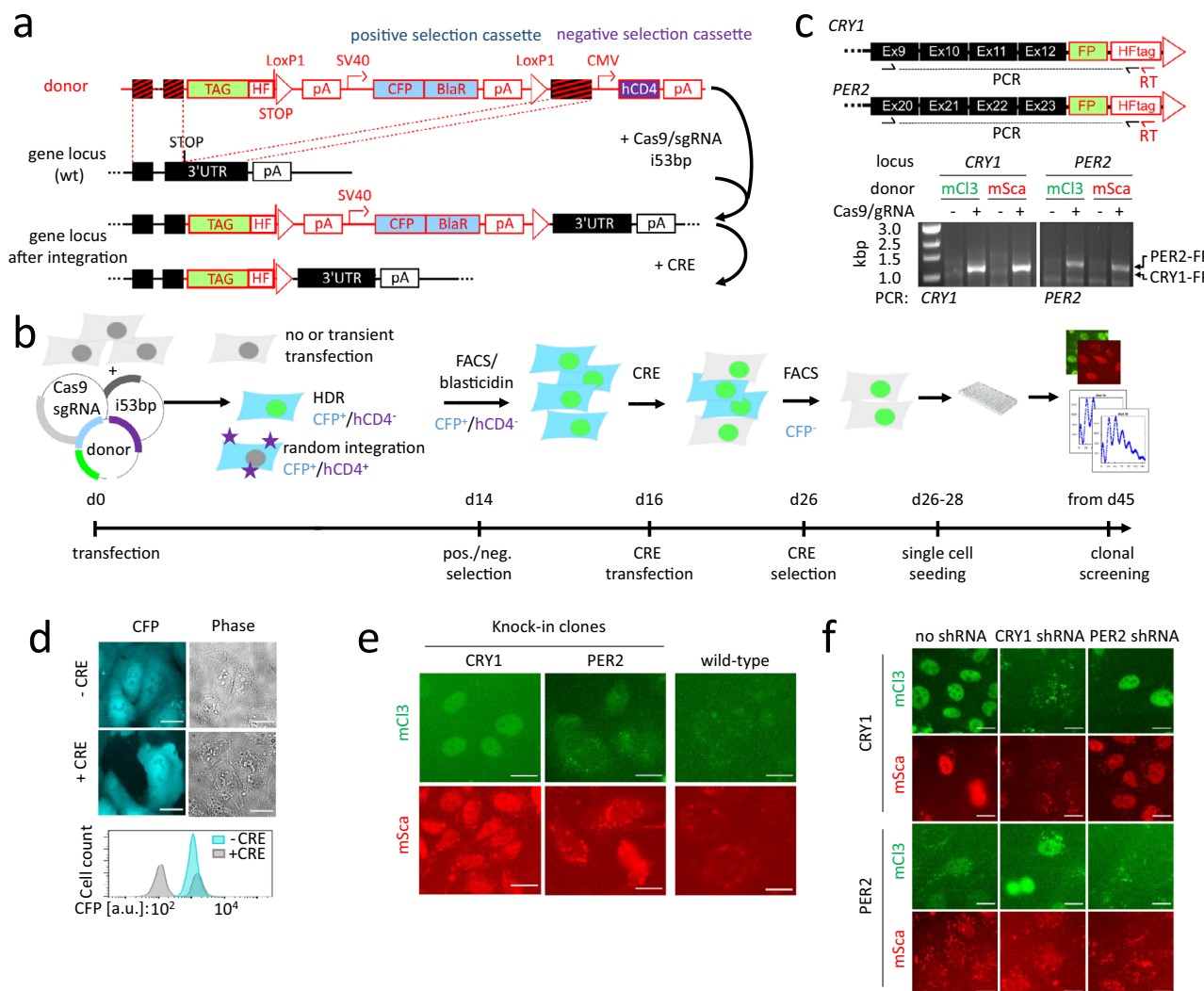

**Fig. 1 CRISPR/Cas9-mediated generation of clock protein knock-in reporter cells. a** Donor plasmid design and genome editing strategy: The tag (e.g. fluorescent protein) to be integrated and a floxed positive selection cassette (cyan fluorescent protein (CFP) + blasticidin-resistance (BlaR)) are flanked by arms, which are homologous to the genomic target region. When Cas9/single guide RNA(sgRNA)-mediated DNA double strand breaks are repaired by HDR, tag, and positive selection marker are integrated into the target region. The negative selection cassette (hCD4, a cell surface protein exclusively expressed on immune cells) is only integrated into the genome by unwanted random integration of the whole donor plasmid. **b** Selection strategy. Cells are transfected with Cas9, sgRNA, i53bp (see text), and donor plasmid. Stable transfectants are selected by blasticidin selection and fluorescence-activated cell sorting (FACS) of CFP positive cells (blue), while unwanted hCD4 positive cells (purple stars) are depleted. Subsequently, cells are transiently transfected with Cre recombinase (CRE) to remove the positive selection cassette from the genomic locus, and only CFP negative cells (gray) are clonally expanded and screened for successful knock-in. **c** Chimeric mRNA was detected in selected batch cultures by reverse transcription-polymerase chain reaction (RT-PCR) using a RT- and a reverse primer specific to the insertion and a gene-specific forward primer. **d** Loss of CFP expression after removal of the positive selection cassette monitored by microscopy and flow cytometry. The gating strategy used FSC and SSC signals to gate out doublets and debris. **e** Fluorescence microscopy images of successful knock-in clones. **f** Indicated knock-in cells were either left untreated or transduced with shRNA targeting either *CRY1* or *PER2*. Images were acquired 10 h after synchronization. Corresponding differential interference contrast (DIC) images are shown in Supplementary Fig. 3a. Scale bars: 20 µm. mCl3 mClover3, mSca mScarlet-I, FP fluorescent protein, pA polyadenylation signal, SV40 simian virus-40 promoter, CMV cytomegalovirus promoter, HF His-tag/FLAG-tag, shRNA short hairpin RNA, UTR untranslated region, Ex exon, FSC forward scatter, SSC sideward scatter. Source data are provided as a Source Data file.

codon sequence. The donor vector contained the coding sequence of mScarlet-I flanked by *PER2* sequences (~800 bp) homologous to those directly upstream and downstream of the STOP codon. A C-terminal 6xHis/FLAG-tag (HF-tag) and a new STOP codon for the fusion protein followed downstream of mScarlet-I (Fig. 1a).

Due to the general low efficiency of HDR, several strategies have been developed to enrich for cells with successful HDR by the use of co-expressed marker proteins, such as fluorescent proteins or an antibiotic resistance. A drawback of most methods is that they have limits for editing of low copy number and/or

transiently (e.g. rhythmically) expressed genes: because marker expression strength correlates with expression of the target gene, correctly edited cells may be co-depleted during 'enrichment' steps. To circumvent these limitations for the low copy number and transiently expressed *PER2* (estimated protein molecules between 0 and 10.000 per cell[19]), we placed the selection marker (a self-cleaving CFP-BlaR fusion protein) in a separate expression cassette downstream of the FP-STOP-codon.

This strategy enables enrichment of cells with genomic integration of the donor by elimination of non- and transiently

transfected cells. To further seperate cells that have integrated the positive selection cassette via HDR at the correct genomic locus from those that have randomly integrated the selection cassette anywhere in the genome, we placed a negative selection cassette into the donor vector outside the homologous arms. This cassette expresses hCD4-extracellular domain and will only be integrated into the genome if the vector is randomly integrated, but not upon HDR. Hence, cells with random integration events can be efficiently eliminated by fluorescence-activated cell sorting (FACS). Finally, flanking LoxP sites allow for removal of the positive selection cassette by transfection with a CRE expression plasmid, thereby restoring the endogenous 3′-UTR - including potential posttranscriptional regulation sites - to the transcript (Fig. 1a).

To summarize, we designed a donor vector with essentially four features: (i) Homology arms plus FP for HDR-mediated editing of target gene. (ii) A positive selection cassette with an independent promoter and poly(A) site driving marker gene expression to select for cells with genomic donor vector integration. Thereby, low or transient expression does not interfere with the selection process. (iii) A negative selection cassette placed outside the homology regions to allow for depletion of cells with random integration of the donor vector. (iv) LoxP sites for removal of the positive selection cassette by transient CRE activity, restoring the genomic locus to essentially an endogenous constitution.

**Generation of knock-in cells.** To create knock-in U-2 OS cells (a human osteosarcoma cell line, which is widely used as a model cell line in the circadian field[40–44], see also Supplementary Note 3) expressing PER2 or CRY1 C-terminally fused to fluorescence proteins from their endogenous promoters, we transfected U-2 OS wild-type cells with spCas9/sgRNA expression plasmid(s), the donor plasmid, and a plasmid expressing an inhibitor of the p53-binding protein, which enhances HDR efficiency by suppressing NHEJ[45] (Fig. 1b). Depending on predetermined sgRNA efficiency (not shown), either a single sgRNA (CRY1) or a mixture of different sgRNAs targeting the same region (PER2) were used.

Two weeks after transfection, the desired cell population was enriched by FACS for CFP⁺/hCD4⁻ cells. The selected cell populations contained chimeric mRNA (target gene and fluorescence tag) (Fig. 1c), indicating that a substantial number of cells of the population had undergone the intended integration events. Subsequent elimination of the floxed positive selection cassette upon transient transfection with CRE recombinase expression plasmid (Fig. 1a) was monitored by the loss of CFP fluorescence (Fig. 1d). Finally, single cell clones were generated from the FACS-selected CFP-negative population, and clonal colonies were inspected using fluorescence microscopy.

In total, 13 out of 31 examined potential CRY1 knock-in clones showed a nuclear fluorescence consistent with data from CRY1 overexpression[13] (Fig. 1e and Supplementary Fig. 1a). For PER2, 7 out of 33 examined clones exhibited more diffuse fluorescence patterns exceeding auto-fluorescence (Fig. 1e and Supplementary Fig. 1b). Fluorescence signals of the other clones were not distinguishable from wild-type cells that served as negative controls (compare Fig. 1e, right panel). Notably, overall fluorescence signals were very low. In addition to the diffuse fluorescence, many cells display punctate fluorescence around the nucleus. Because these signals were detectable in different fluorescence channels (in contrast to the specific signals), exhibit a much shorter fluorescence lifetime compared to specific signals ($1820 \pm 90$ ps vs. $2830 \pm 120$ ps) and were also present in wild-type cells (Supplementary Fig. 1c–f), we classified these signals as cellular auto-fluorescence. In many cases, such punctate auto-fluorescence signals were of stronger

intensity than the actual diffuse signals from the introduced fluorophores, thus impeding quantification of fluorescence especially in the cytoplasm.

Strikingly, all five tested CRY1 knock-in clones with clear nuclear fluorescence pattern were positive for the chimeric mRNA (Supplementary Fig. 1g), while one tested clone with a different fluorescence pattern (Supplementary Fig. 1a) was not. Out of seven tested fluorescence positive PER2 knock-in clones, three were positive for the corresponding chimeric mRNA (Supplementary Fig. 1g). Sanger sequencing confirmed the identity of the PCR products.

Next, we examined the targeted genomic loci of these clones. To this end, we amplified the locus from genomic DNA of several clones using PCR primers that both bound outside the employed homology region ('out-out-PCR', Supplementary Fig. 1h). For most tested clones, we detected several PCR products: a major product corresponding to the size of the wild-type allele amplicon and minor products, including one with the expected size range of a knock-in allele amplicon, indicating a mono-allelic integration. Additional products are probably due to formation of heteroduplexes between wild-type and knock-in PCR products. Sanger sequencing of all tested knock-in alleles revealed exact matches with the predicted sequences for a successful knock-in, and confirmed a precise excision of the positive selection cassette (Supplementary Fig. 2a–d). For some tested clones, only the knock-in product was detected, which suggests a knock-in at both alleles. To further determine genomic copy numbers of the fluorophores (mClover3, mScarlet-I) relative to the targeted genes, we performed digital droplet PCR (ddPCR). For the tested CRY1-mScarlet clone, we found a ratio of fluorophore to target gene of 1:2 indicating a mono-allelic knock-in, consistent with the result of the 'out-out-PCR' (Supplementary Fig. 1i). Surprisingly, in one PER2-mClover3 clone, the fluorophore to target gene ratio was 1:3, suggesting a third copy of PER2 in this clone. Indeed, we obtained a ratio of PER2:CRY1 of 3:2 for this clone. For the CRY1-mClover3 clone, the ratio of fluorophore to target gene was 1:1, confirming the biallelic knock-in also found by 'out-out-PCR'. In contrast, the PER2-mScarlet-I clone yielded a ratio of 1:2, indicating a knock-in on only one allele. Of note, these obtained ratios also indicate that no further (randomly integrated) copies of the fluorophore were present in the genome of any tested clone.

We also sequenced the second alleles of the mono-allelic integration clones. While the coding sequence was wild-type in two cases, Cas9 induced insertions or deletions for the other clones, resulting in alterations of up to 18 amino acids at the C-terminus. For two clones (both PER2-mScarlet-I), larger deletions (~564 bp and 3878 bp, respectively) were observed at the second allele (Supplementary Fig. 2a–d), leading to deletion of the last exon's coding region. This also explained the absence of the wild-type PCR band in the latter one, as the binding region of the forward primer was missing at this allele.

Cas9 has the known potential to cut at multiple off-target sites, thereby introducing further unwanted alterations of the genome (mostly insertions or deletions, i.e. indels)[46]. Since indels in exonic regions are likely to be detrimental for protein functions, we performed whole exome sequencing (WES) of several knock-in clones and searched for alterations at predicted Cas9 off-target sites. While we detected indels at the wild-type alleles of the on-target site, as also seen by Sanger sequencing (Supplementary Fig. 2), we did not find evidence for Cas9-induced indel formation near 1,576 predicted potential exonic off-target sites. In addition, the WES data provided no evidence for random integration of any donor plasmid sequence at exonic regions, which may disrupt a gene's function at the integration site (see Supplementary Note 1 for more details).

Finally, we wanted to exclude that the observed fluorescence signals originate from any other source than the PER2- or CRY1-fusion proteins. To this end, we transduced the knock-in clones with shRNA targeting *PER2* or *CRY1* mRNA and recorded fluorescence over 24 h (Fig. 1f and Supplementary Fig. 3). In *CRY1* knock-in cells treated with shRNA against *CRY1* fluorescence signals were reduced to below 20% (Supplementary Fig. 3b, c, f), while cells left untreated or treated with shRNA against *PER2* showed robust nuclear fluorescence signals (red or green). Similarly, transient fluorescence signals in *PER2* knock-in cells were even enhanced upon knockdown of *CRY1* (probably due to reduced repression of *PER2* transcription). In contrast, mean fluorescence of cells transduced with an anti-*PER2*-shRNA was not distinguishable from auto-fluorescence of wild-type cells at any time point (Supplementary Fig. 3d–f). Together, this strongly indicates that essentially all of the observed fluorescence is due to proteins translated from the same mRNA as CRY1 or PER2 proteins, and thus fluorescence originates exclusively from the targeted fusion proteins. This is in line with the results from the copy number analysis by ddPCR, which indicated the absence of additional fluorophore copies in the knock-in clones (Supplementary Fig. 1i).

Combining fluorescence, PCR and sequencing data, we identified between 5 and 56% (median 19%) of the initially screened clones as successful knock-in clones (Supplementary Fig. 1j). All tested clones had fluorescent proteins exclusively present at the targeted sites and showed no off-target insertions or deletions at other exonic regions. For each knock-in (*PER2*-mClover3, *PER2*-mScarlet-I, *CRY1*-mClover3, and *CRY1*-mScarlet-I), we chose one clone for further experiments.

**Clock protein knock-in cells possess an intact circadian clock**. Knocking in fluorescent protein tags into *PER2* and *CRY1* genomic loci allows studying the endogenous clock protein's dynamics in living cells, if the knock-in does not affect the functionality of the molecular oscillator. To test this, we monitored *Bmal1*-promoter driven luciferase rhythms over several days in five selected clones (mono-allelic *PER2*-mClover3, two mono-allelic *PER2*-mScarlet-I, bi-allelic *CRY1*-mClover3 and mono-allelic *CRY1*-mScarlet-I). All clones showed robust circadian oscillations with amplitudes and periods similar to wild-type U-2 OS cells (Supplementary Fig. 4a–d). This was expected for C-terminally tagged PER2, since homozygous *Per2*-luciferase or *Per2*-Venus knock-in mice show normal circadian locomotor behavior[16,34]. Adding a C-terminal tag to CRY1 could in principle lead to a hypomorphic allele with altered functionality of the corresponding fusion protein, which might go undetected in cells with only one allele carrying the knock-in and the other essentially being wild-type. This is unlikely, however, since (i) various C-terminal tags did not alter CRY1's ability to repress CLOCK:BMAL1 transactivational activity (Supplementary Fig. 4e) and (ii) the *CRY1*-mClover3 clone with a bi-allelic knock-in showed essentially normal circadian dynamics (no period shortening as expected for hypomorphic alleles). In summary, the molecular clock in the tested knock-in cells was still functional and there was no indication that the fusion proteins represent non-functional variants.

**Protein dynamics of CRY1 and PER2**. In the past, predominantly biochemistry experiments (with cell populations) suggested that both PERs and CRYs first accumulate in the cytoplasm, while their nuclear abundance shows circadian rhythms with peak levels in peripheral tissues at CT16-20[17,26,31]. To test whether and to what extent this is also true in individual living cells, we monitored fluorescence in synchronized knock-in cells at regular 1-h intervals over the course of 3 days.

In contrast to our expectations, the fluorescence of CRY1 fusion proteins was exclusively observed in the nucleus at any given time point (Fig. 2a, b and Supplementary Movies 1–2) with fluorescence levels observed in the cytoplasm indistinguishable from background levels in wild-type U-2 OS cells (Supplementary Fig. 4f). This indicates that the majority of CRY1 is predominantly in the nucleus irrespective of time of day. The nuclear signal intensity was well above background at all time-points in almost all cells and oscillated with a circadian period in the majority of cells. In contrast to CRY1, nuclear fluorescence signals of PER2 fusion proteins were more transient and detected for only 8–12 consecutive hours in an individual cell (Fig. 2c, d and Supplementary Movies 3–4), resulting in circadian rhythms of PER2 nuclear signal consistent with previous reports from *Per2*-Venus knock-in mice[16]. Prior to nuclear accumulation, a weak cytoplasmic PER2-FP signal was detectable in some cells; however, a reliable discrimination from auto-fluorescence and background signals was not possible (Supplementary Fig 4g). In summary, we concluded that (i) the circadian clock was still intact in our knock-in clones, (ii) both CRY1 and PER2 fusion protein levels oscillate in a manner consistent with their well-established circadian regulation, and (iii) PER2 and CRY1 expression dynamics can be monitored in the nuclei of single cells.

To exclude that the observed expression patterns are specific to U-2 OS cells, we also knocked-in dClover2 and mScarlet-I into the *PER2* and *CRY1* locus of the human colon epithelia cell line HCT-116 (another circadian model cell line[47–49]) and monitored fluorescence over the course of two days (Supplementary Fig. 5a, b). The spatiotemporal fluorescence patterns essentially recapitulated those seen in the U-2 OS cells: CRY1 was detectable almost exclusively in the nucleus over the whole circadian cycle, while nuclear PER2 fusion protein signal was detectable for less than 12 consecutive hours, indicating that the dynamics of PER2 and CRY1 are similar in human cells.

**CRY1 is phase-delayed compared to PER2**. To obtain a more quantitative picture of the spatiotemporal dynamics of PER2 and CRY1, we tracked nuclear fluorescence of ~20 individual cells of each knock-in clone over 3 days (Fig. 2e). We used MetaCycle[50] to analyze the time series for the presence of circadian rhythms. Both CRY1-mScarlet-I and CRY1-mClover3 showed significant circadian rhythmicity of nuclear abundance in almost all cells with average periods of $24.7 \pm 2.0$ h and $25.6 \pm 1.6$ h (mean ± SD), respectively (Fig. 2f, g). Circadian rhythms of nuclear fluorescence were also observed for the majority of *PER2*-mScarlet-I and *PER2*-mClover3 knock-in cells with average periods of $25.9 \pm 2.0$ h and $25.3 \pm 1.7$ h, respectively (Fig. 2f, g). The average relative amplitudes of rhythms of PER2-fusion protein nuclear abundance were twice as high as those of CRY1-fusion proteins (Fig. 2h). When comparing the average phases of PER2 and CRY1 protein rhythms in the nucleus, CRY1 fusion proteins appear to be phase-delayed relative to corresponding PER2 proteins by more than 3 h (Fig. 2i). However, at this stage it remained unclear whether this reflects phase differences in individual cells or resulted from clonal variation. We also observed a minor phase advance of the Scarlet-fusion proteins compared to the corresponding mClover3 fusion proteins of ~1 h (Fig. 2i), which – in addition to clonal variation – possibly reflects differences in maturation time.

**Generation of double knock-in cells**. To test whether delayed nuclear CRY1 accumulation is due to variability between individual cells or whether it is indeed a feature of the circadian oscillator, we generated knock-in cells expressing both CRY1 and PER2 as fluorescence tagged fusion proteins in different colors. To this end, we used *PER2*-mClover3 and *PER2*-mScarlet-I

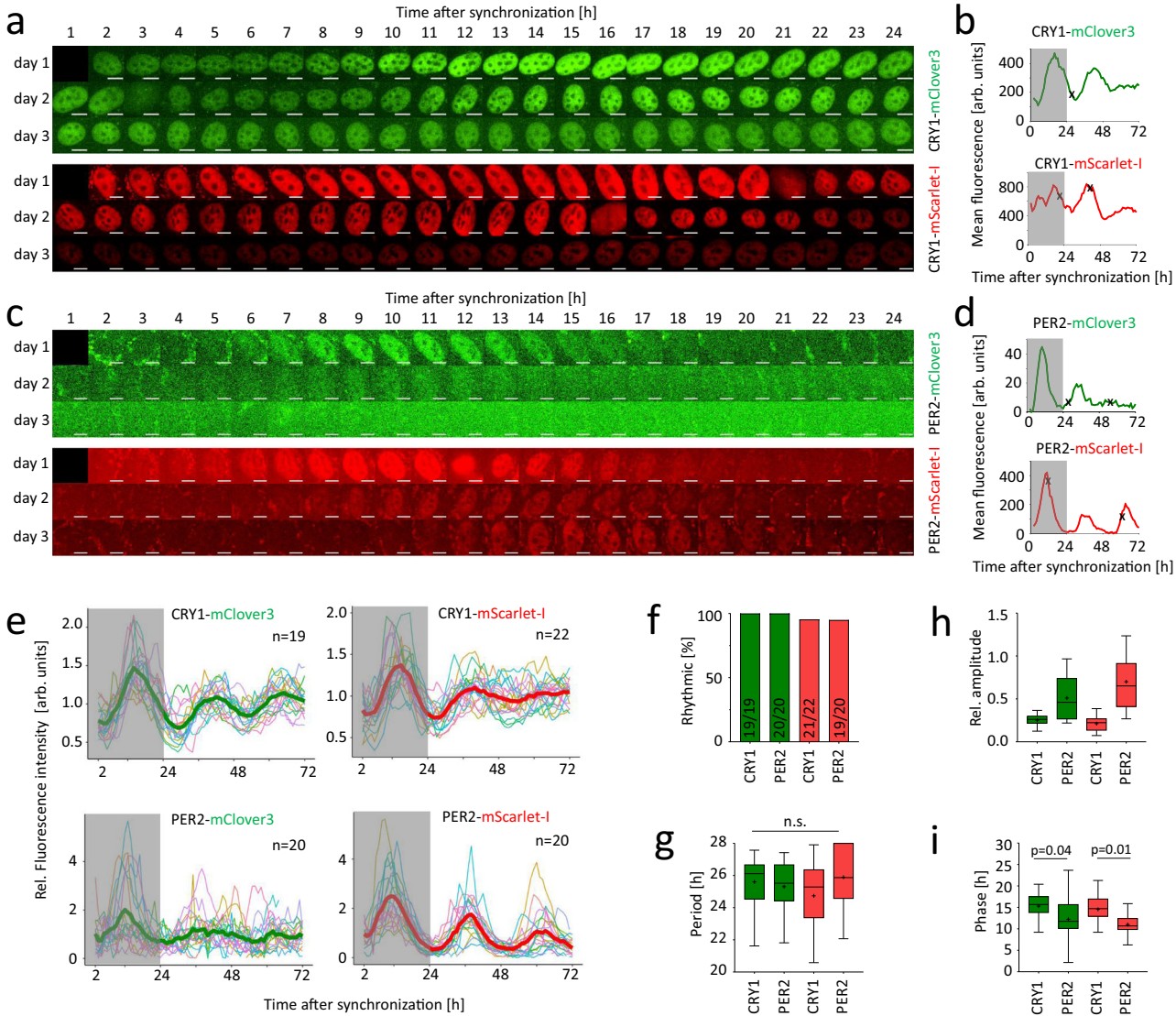

**Fig. 2 PER2- and CRY1-fusion protein oscillate in single knock-in cells. a**, **c** Montage of fluorescence microscopy images of selected individual U-2 OS single knock-in cells' nuclei over the course of 3 days after synchronization. **b**, **d** Mean of nuclear fluorescence intensity (background-subtracted) quantified from (**a**) and (**c**). Cell division marked by (x). **e** Time series of normalized mean nuclear fluorescence in individual knock-in cells with average signal overlaid in bold. f Percentage of significantly rhythmic time series from (**e**). **g**–**i** Extracted rhythm parameters of significantly rhythmic single-cell time series from **e** (n = individual cells as stated in **f**). p-values: one-way ANOVA, two-sided. Scale bar: 10 μm. Boxplots: box: interquartile range, center: median, whiskers: minimum to maximum, mean is marked with (+). In **b**, **d** and **e**, the first 24 hours after synchronization (possible acute response to dexamethasone) are highlighted by a gray box. Source data are provided as a Source Data file.

mono-allelic knock-in cells to generate double knock-in cells by integration of the complementary fluorophore into the *CRY1* locus as described above (Fig. 1a, b). After positive and negative selection and CRE-mediated excision of the selection cassette, single clones were screened by microscopy. Again, high proportions of the inspected clones (12 out of 14 (86%) for *CRY1*-mScarlet-I knock-in and 11 out of 19 (58%) for *CRY1*-mClover3 knock-in) showed similar nuclear pattern as seen in the *CRY1* single knock-in clones (Fig. 3a, b and Supplementary Fig. 6a). Three clones of each were selected, and RT-PCR and PCR/ sequencing of the *CRY1* locus confirmed successful knock-in for all clones (Supplementary Fig. 6b, c and Supplementary Fig. 2a, b). We further analyzed the genome of two *PER2*-mScarlet-I/ mClover3 double knock-in clones (#4 and #6) by ddPCR and WES. For both, we obtained a fluorophore-to-transgene ratios of 1:2, indicating a specific mono-allelic knock-in at both target

genes with no further copies of the fluorophore in the genome (Supplementary Fig. 6d). As for the single knock-in cells, we did not find strong evidence for Cas9-mediated indel formation or random integration of the donor vector at exonic sites (for details, see Supplementary Note 1). Live-cell fluorescence microscopy revealed, as expected, that the spatial fluorescence patterns of PER2 and CRY1 fusion proteins substantially overlap in the nucleus (Fig. 3a). Furthermore, circadian rhythms were still intact in those cells as confirmed by bioluminescence imaging using a *Bmal1*-luciferase reporter (Supplementary Fig. 6e–h).

**CRY1 is phase-delayed compared to PER2 also in individual cells.** To quantify the temporal relationship of nuclear CRY1 and PER2 protein expression, we synchronized *CRY1*-mClover3/ *PER2*-mScarlet-I double knock-in cells and determined nuclear fluorescence intensity of 50 individual cells over the course of

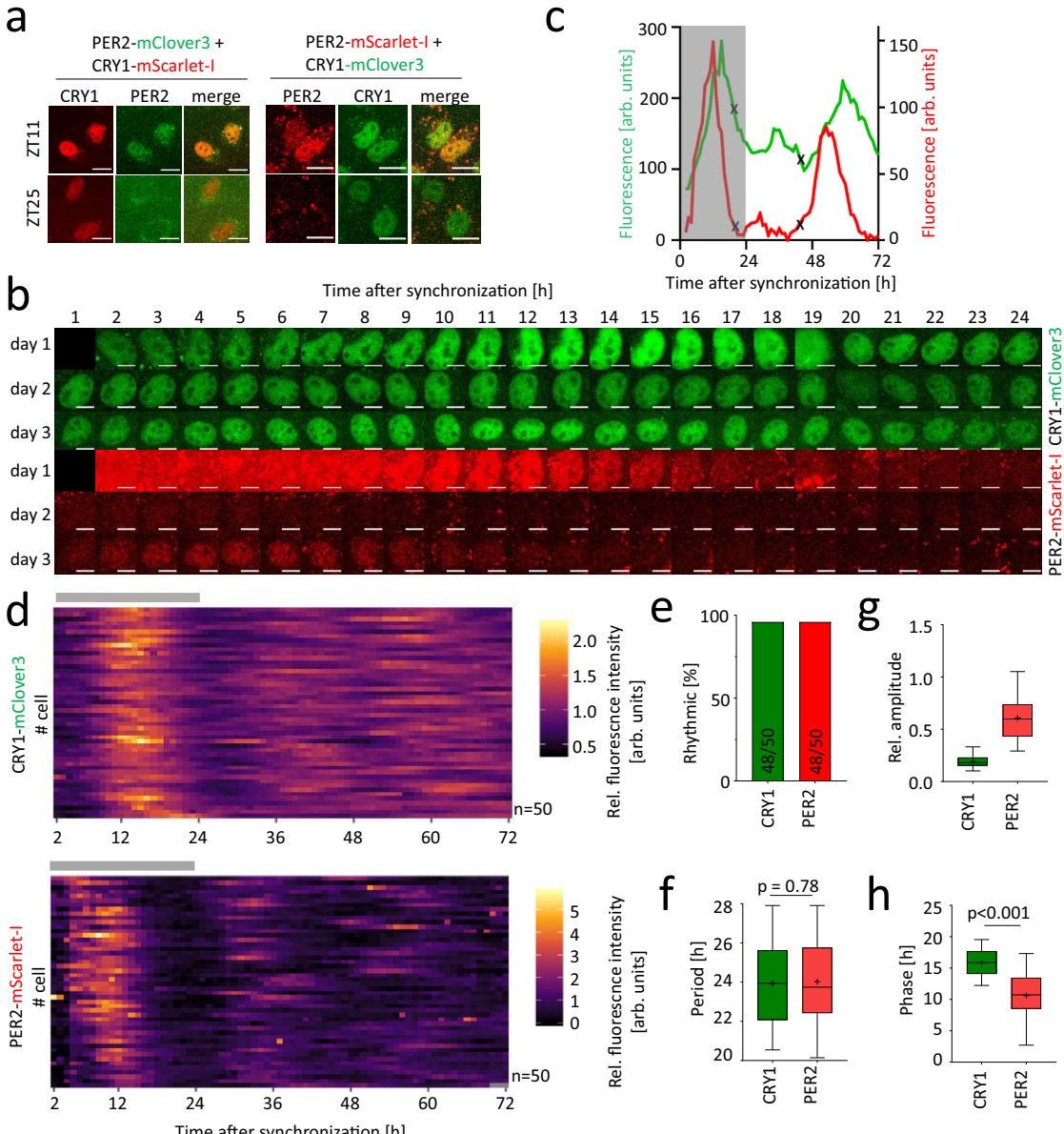

**Fig. 3 Simultaneous visualization of PER2- and CRY1-fusion protein oscillations in double knock-in cells. a** PER2- and CRY1-fusion protein oscillation in individual double knock-in cells. Fluorescence images of selected double-knock in clones at different times after synchronization. **b** Montage of bicolor fluorescence microscopy images of an individual U-2 OS double-knock-in cell's nucleus over the course of 3 days after synchronization. **c** Mean nuclear fluorescence intensity (background-subtracted) quantified from (**b**). Cell division marked by (x). **d** Time series of normalized mean nuclear fluorescence in individual double knock-in cells. **e** Percentage of significantly rhythmic time series from **d** (n = individual cells as stated in **f**). **f–h** Extracted rhythm parameters of significantly rhythmic single-cell time series from **d**. p-values: Mann-Whitney-U test, two-sided, for **h**, p-value = 2*10$^{-15}$. Boxplots: box: interquartile range, center: median, whiskers: minimum to maximum, mean is marked by (+). Scale bar: 10 μm. The first 24 hours after synchronization (possible acute response to dexamethasone) are highlighted by a gray box (**c**) or bar (**d**). ZT = zeitgeber time. Source data are provided as a Source Data file.

3 days (Fig. 3b–d and Supplementary Movie 5). Again, almost all cells (>95%) displayed significant rhythmicity of CRY1 and PER2 levels with average periods of 24.0 ± 2.3 h for both proteins and a ~3-fold higher relative amplitude of PER2 rhythms (Fig. 3e–g).

Very similar results were obtained from HCT-116 *CRY1*-mScarlet-I/*PER2*-dClover2 double knock-in cells (Supplementary Fig. 5c–f). Although the amplitude of CRY1 nuclear expression rhythms in these cells was higher than in U-2 OS cells, it was still significantly lower than that of PER2 (Supplementary Fig. 5e, f). Overall, data from the double knock-in HCT-116 cells recapitulated well the results obtained with U-2 OS cells

As seen in the single knock-in cells, mean CRY1-mClover3 nuclear fluorescence signal followed that of PER2-mScarlet with a delay of ~5 h (Fig. 3c, h). In individual cells, the median phase difference between PER2 and CRY1 nuclear abundance rhythms was 5.4 h (Fig. 4a). To test whether this phase difference is dominated by differential responses of PER2 and CRY1 expression to dexamethasone synchronization, we reanalyzed the time series starting from 26 h post treatment, thus omitting data from the first circadian cycle. Again, the phase of CRY1 rhythmicity was still delayed by 4.9 h compared to PER2 rhythms (Fig. 4b) indicating that the observed delay was not due to acute

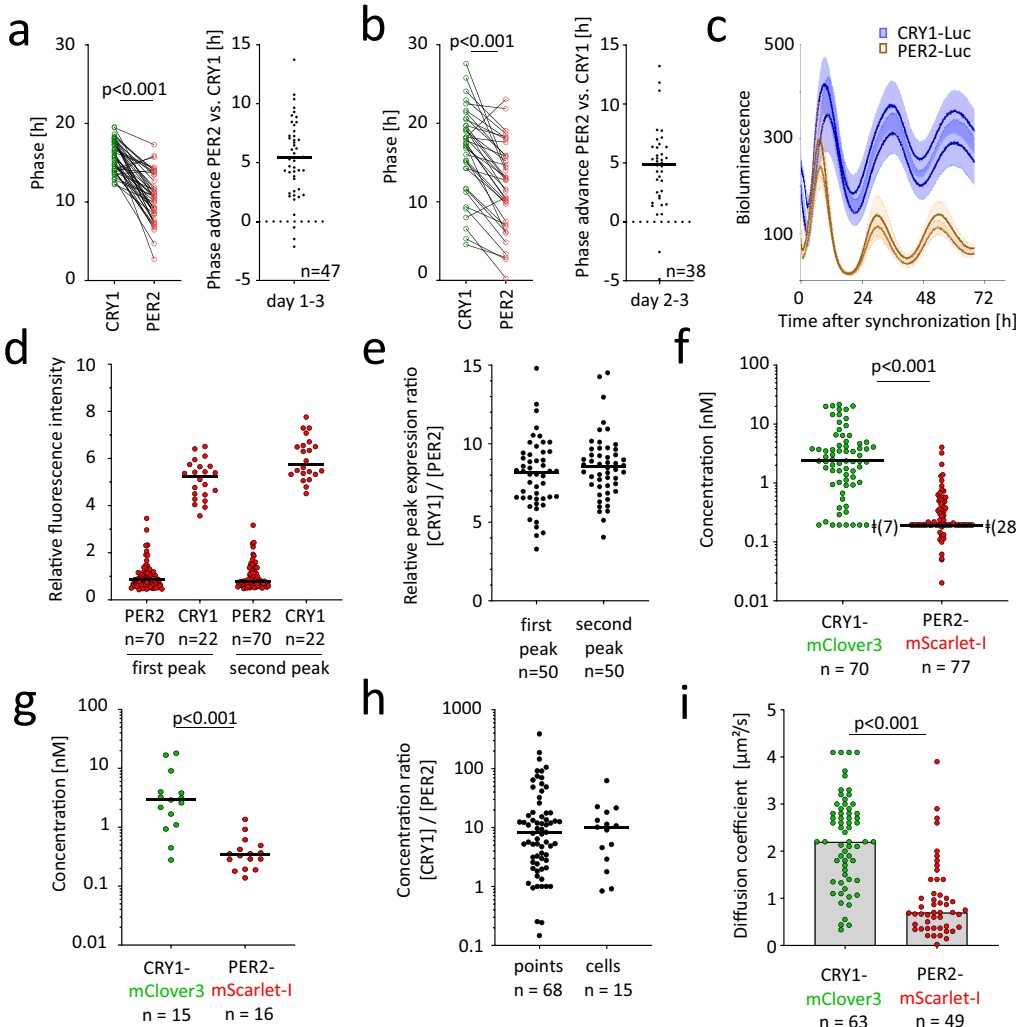

**Fig. 4 Nuclear CRY1 peaks later and is much more abundant than PER2. a, b** Analysis of phase difference between CRY1 and PER2 nuclear accumulation in individual double knock in cells. Phases were calculated either including (**a**) or excluding (**b**) the first 24 hours of the three days' time series. **c** Live-cell bioluminescence recordings of knock-in cells expressing either CRY1-luciferase or PER2-luciferase fusion proteins. Depicted are mean ± SD of 6 individual traces from 2 independent experiments with 2 clones of each knock-in. Parameter analysis in Supplementary Fig. 7a. **d** Relative nuclear peak intensity of fluorescence in cells expressing either PER2- or CRY1-mScarlet fusion protein. **e** Ratio of normalized expression of CRY1-mClover3 versus PER2-mScarlet-I in individual cells expressing both fusion proteins. **f–i** Protein concentrations and diffusion coefficients of CRY1 and PER2 fusion proteins in double knock-in cells determined by fluorescence correlation spectroscopy (FCS) in up to five measurement areas of individual cells' nuclei at the time of estimated PER2 peak expression. **f** Concentration in single measurement areas. (‡) marks values not exceeding background signal, which is set to estimated faithful limit of detection (numbers of points in brackets, see Materials and Methods section for more information). **g** Mean concentration per cell from data points shown in (**f**). **h** Ratio of CRY1-mClover3 versus PER2-mScarlet-I concentration per measurement area (left) and per individual cell (right). i Diffusion coefficients in single measurement areas. Lines in scatter plots depict median. SKI: single knock-in. DKI: double knock-in. p-values: Wilcoxon signed-pair rank test, two-sided (**a** and **b**, p-values are $6.1 \times 10^{-13}$ and $1.3 \times 10^{-8}$, respectively), Mann-Whitney test, two-sided (**f, g, i**, p-values are $<10^{-15}$, $1.1 \times 10^{-5}$, and $5.5 \times 10^{-12}$, respectively). Numbers (n) refer to individual cells for **a, b, d, e, g** and **h** (right panel), and individual imaging areas for **f, h** (left panel) and **i**. Source data are provided as a Source Data file.

dexamethasone effects. Similar results were obtained with U-2 OS cells synchronized by a cold temperature pulse or medium exchange (Supplementary Fig. 7a, b) as well as with double knock-in HCT-116 cells (Supplementary Fig. 7c), indicating that the observed phase differences between nuclear PER2 and CRY1 does not depend on synchronization agent or cell type.

To investigate whether the PER2-CRY1 phase difference is specific for nuclear accumulation or whether it is a feature of their whole-cell expression dynamics, we knocked in firefly luciferase into the *CRY1* or *PER2* loci of U-2 OS cells using the same strategy and recorded luminescence of three individual clones over the course of three days. (Reliable quantification of

whole-cell fluorescence signals was impossible, because of the rather low fluorescence and high auto-fluorescence signals in the cytoplasm of FP-reporter cells). As observed for fluorescence fusion proteins, the phase of CRY1-LUC expression rhythms was delayed by ~5 h compared to that of PER2-LUC (Fig. 4c and Supplementary Fig. 8).

**CRY1 is much more abundant than PER2 in the nucleus of U-2 OS cells.** Mass spectrometry data from mouse liver suggested that CRY1 protein peak levels are higher than those of PER2[19]. Since we have tagged CRY1 and PER2 with the same fluorescent proteins, we were able to quantitatively compare the expression

intensities of both proteins in U-2 OS cell nuclei. To this end, we quantified peak fluorescence levels, from *CRY1*-mScarlet-I knock-in cells, from *PER2*-mScarlet-I knock-in cells, and from double knock-in cells expressing PER2-mScarlet-I from one allele (and also CRY1-mClover3 from one allele). Assuming that the distribution of peak expression levels are roughly similar between the different knock-in cell lines, we estimated that average peak expression of CRY1 is more than five times higher than that of PER2 (Fig. 4d). In a similar manner, U-2 OS *CRY1*-luciferase knock-in cells gave rise to much higher signals in comparison to PER2-luciferase knock-in cells on a population level, indicating that these abundance level differences are not confined to the nucleus (Fig. 4c).

To compare peak intensities between PER2 and CRY1 on single-cell level, we took advantage of the fact that we have generated different cell lines expressing CRY1 coupled to either mScarlet-I or mClover3. Again assuming that the distribution of CRY1 peak expression levels is roughly similar among the different knock-in cell lines, we compared the signal intensities of CRY1-mScarlet-I and CRY1-mClover3 at the peak of circadian expression. This allowed us to estimate the relative amount of PER2-mScarlet-I and CRY1-mClover3 in individual double-knock-in cells. Similar to the population level, the peak amount of CRY1 surpassed that of PER2 in the same cells by 8.0 ± 2.2 fold for the first peak and by 8.5 ± 2.1 fold for the second (mean ± standard deviation, Fig. 4e), suggesting that CRY1 protein is present at much higher levels than PER2 protein in the nucleus of U-2 OS cells.

To confirm the obtained nuclear CRY1:PER2 ratio by an independent technique, we applied fluorescence correlation spectroscopy (FCS), which allows estimating the molecule concentration by measuring diffusion events through a small confocal volume[51,52]. Using FCS, we determined the molecular concentrations of PER2-mScarlet-I and CRY1-mClover3 in nuclei of double-knock in cells 34–36 h after dexamethasone synchronization (2nd peak of PER2 expression) and obtained a mean nuclear concentration of 4.7 nM for CRY1-mClover3 and of 0.42 nM for PER2-mScarlet-I (Fig. 4f, g, medians of 2.9 nM and 0.34 nM, respectively). Note, that these values refer to the contribution of only one (knock-in) allele. Thus, the median ratio of CRY1 to PER2 nuclear abundance was about 10-fold, (Fig. 4h), again indicating a much higher nuclear abundance of CRY1 in our cells. Interestingly, diffusion constants were higher for CRY1-mClover3 (2.3 ± 0.9 μm²/s) than for PER2-mScarlet-I (1.0 ± 0.8 μm²/s), indicating that a substantial part of CRY1 is not integrated into large protein complexes (Fig. 4i).

**Insights from mathematically modeling nuclear PER2 and CRY1 dynamics**. Inspired by these unexpected observations, we wanted to better understand the underlying regulatory mechanisms. First, why are the absolute levels of nuclear CRY1 so much higher than those of nuclear PER2, given the consensus model that PER2 and CRY1 enter the nucleus simultaneously and co-exist mainly within large complexes? Second, why does PER2 oscillate with higher amplitudes than CRY1? Finally, what mechanism might control the phase difference between nuclear PER2 and nuclear CRY1? To gain insight into the implications of our experimental observations in both a conceptual and a quantitative manner, we constructed a mathematical model of the mammalian circadian oscillator based on our data (Supplementary Fig. 9) emphasizing the nuclear PER2 and CRY1 dynamics (Fig. 5a). To this end, we took a published model[53] and modified it at three main stages: (i) We simplified the PER2-CRY1 loop by removing the PER2 phosphorylation module. (ii) We added a dissociation event of the nuclear PER2:CRY1 complex to the

respective monomers. (iii) We did not assume degradation of the cytoplasmatic and nuclear PER2:CRY1 complexes, but rather assumed degradation of the monomers after dissociation of the complex. The details of the resulting model are presented in Supplementary Fig. 9 and the corresponding modified equations in Supplementary Note 2. Variables and parameters are given in Supplementary Tables 6 and 7.

We explored the parameter space of the PER2:CRY1 loop and found a set of parameters that reproduced our experimental findings, i.e. (i) a circadian period, (ii) a higher abundance of nuclear CRY1 protein compared to PER2, (iii) a larger amplitude of PER2 rhythms, and (iv) a delayed peak of nuclear CRY1 protein compared to PER2 rhythms (Fig. 5b). Our modeling results predict: First, the half-life of nuclear CRY1 is higher than that of nuclear PER2. Second, association of cytoplasmatic PER2 and CRY1 occurs at a higher rate than the dissociation event and, conversely, in the nucleus, dissociation of the PER2:CRY1 complex occurs at a higher rate than the association, thus driving the net PER2:CRY1 flux toward nuclear monomers. These predictions are summarized in Fig. 5a.

This modeling strategy also presented the opportunity to study properties of the underlying oscillating system in a combined experimental-theoretical approach. Because single cell data are notoriously noisy (see Figs. 2e and 5c as well as Source Data with all image-extracted time series data), a direct fitting of gene-regulatory models such as the one presented in this study is difficult. For this reason, we analyzed fundamental oscillator properties (periods and amplitudes) from both the data and model simulations. Given the enormous cell-to-cell variability in critical cellular processes, such as transcription[54], translation[55], degradation[56], and nuclear translocation[57], we applied an ensemble approach: We examined the interdependencies of oscillatory parameters for all significantly rhythmic experimental time series and compared them with model simulations. This approach allowed us to test the following hypothesis: while for linear harmonic oscillators the periods do not depend on the amplitudes, we hypothesized that for circadian (non-linear) oscillators there are specific amplitude-period dependencies as described in the pioneering studies of Georg Duffing[58]. Indeed, analyzing the time series of 87 individual cells, we found a striking positive correlation of PER2 amplitude and period (Fig. 5c). Importantly, our modified model reproduced this dependence (we simulated cell-to-cell variability by randomly changing all transcription, translation, degradation, as well as translocation parameters). Such relationships between the amplitudes and periods of an oscillator are called "twist" and have been discussed previously[59].

## Discussion

The current model of the mammalian circadian oscillator is predominantly based on data from genetics and biochemistry experiments that have been accumulated over more than 20 years. In addition, luciferase reporter technology substantially advanced our knowledge about the dynamics of circadian rhythms in live cells. The cell biology of circadian clocks, however, is still in its infancy mainly due to the lack of suitable reporter technologies that allow the (simultaneous) spatiotemporal quantification of individual clock proteins in living single cells. This is likely due to the fact that even almost a decade after the CRISPR revolution, the generation of knock-in cell lines is still not a standard technique, but a time consuming endeavor with uncertain success. Here, we enrich the existing toolbox by an efficient selection strategy that is particularly useful for genes expressed transiently or at low level, such as those coding for circadian clock proteins. The selection process is independent of target gene expression

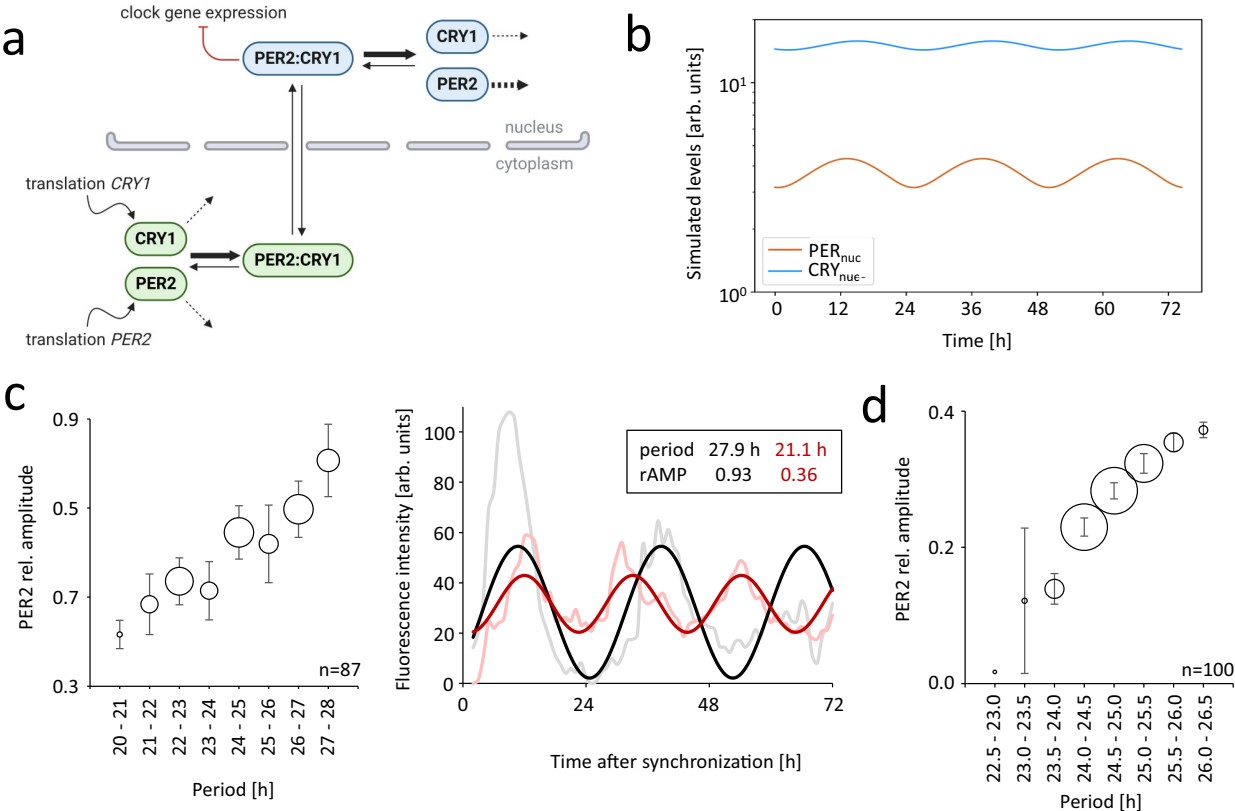

**Fig. 5 A mathematical model of nuclear PER2 and CRY1 dynamics in mammalian circadian clock cells reproduces experimental findings. a** Model of the PER2:CRY1 loop together with the predictions regarding half-lives and association/dissociation events that arise from our simulations (see main text). Dashed black lines represent degradation events; thicker arrows represent reactions that are predicted to occur at a higher rate. The red solid arrow depicts the inhibition of clock gene expression, exerted solely by the PER2:CRY1 nuclear complex. For a full reasoning of the model design and parameter choice, see Supplementary Note 2 and Supplementary Fig. 9. **b** The model exhibits sustained 24.7 h oscillations with nuclear PER2 preceding CRY1 rhythms, and with PER2 oscillating with a higher amplitude and lower magnitude than CRY1, thus reproducing experimental results. The parameter set is given in Supplementary Tab. 7. **c** The amplitude of PER2 rhythms increases with period in knock-in cells with significantly rhythmic oscillations (n = 87 cells, left). Exemplary raw time series and corresponding cosine fits (right) illustrate that long-period rhythms typically display larger amplitudes. The size of the circles depict the number of cells that fall in each bin. Error bars represent SEM. **d** Our mathematical model reproduces the positive correlation of PER2 amplitude and period. We simulated n = 100 "artificial" cells by randomly varying all transcription, translation, degradation, nuclear import and export rates. Parameter values were drawn from a uniform distribution on the interval given by the default parameter value ± 10% (see Supplementary Note 2 for simulation details). The size of the circles depict the number of cells that fall in each bin. Error bars represent SEM. rAMP relative amplitude.

and only leaves a single loxP site in the genome. For achieving high knock-in efficiencies, our approach can be combined with complementary techniques, such as the use of CRISPR/Cpf1 or cell cycle synchronization[60,61].

The resulting fluorescent knock-in cell lines enabled us to visualize endogenously expressed PER2 and CRY1 proteins – two canonical circadian repressors –in single human live cells (U-2 OS and HCT-116 cells; for a discussion on the U-2 OS cell line as a circadian model, see Supplementary Note 3). We are confident that the fusion proteins are functional, because cells with a mono-allelic knock-in of PER2-fusion proteins or a bi-allelic knock-in of CRY1-fusion proteins display normal circadian rhythms, in contrast to *Per2* or *Cry1* knockout cells which are hardly rhythmic[62,63]. This is further supported by reports that homozygous *Per2*-Venus and *Per2*-Luc knock-in mice display normal circadian rhythms, and rhythmically expressed CRY1-EGFP can rescue rhythmic behavior in otherwise arrhythmic *Cry1/2* double knock-out mice[16,25,64]. Therefore, these knock-in cells should constitute reliable tools to study the spatiotemporal dynamics of both proteins within a widely used model of the human circadian oscillator on the single-cell level.

Circadian oscillations occur because of a critical delay in the auto-regulatory negative feedback of PER and CRY proteins on their own transcription. Delayed nuclear accumulation of negative regulators has been discussed to be one mechanism in this context. Indeed, in mammals, nuclear levels of PER and CRY proteins seem to mutually depend on the presence of the other protein family members[12]. In *Drosophila*, the analogs of PER and CRY, dPER and dTIM, are reported to first accumulate in the cytoplasm for several hours and then translocate into the nucleus in a switch-like event[14].

So far, our knowledge about CRY1 protein subcellular localization dynamics mostly results from western blot analysis of cytoplasmic and nuclear fractions. However, the picture is not clear: On the one hand, several studies suggested that CRY1 nuclear abundance is mainly restricted to the repressive phase (early and late), resulting in an rather sharp peak of nuclear CRY1 protein between CT12 and CT21 in peripheral tissues[43,65]. On the other hand, other studies reported the presence of nuclear CRY1 protein across the whole circadian cycle[4,12,17,27,29,66,67]. Different experimental protocols, cell types (species and tissue), and synchronization status may have contributed to these apparent

contradictory results. In addition, since most published data are generated from analyzing populations of thousands of cells, it is largely unknown, whether observed circadian protein dynamics are indeed cellular properties or rather stem from cellular heterogeneity.

Using our reporter cells, we found three interesting features of circadian clock protein dynamics in human cells that may help to refine the current model of the circadian oscillator. Firstly, CRY1 is in the nucleus at all circadian phases in U-2 OS cells (and also in HCT-116 cells) with no detectable CRY1 in the cytoplasm. Although from our data we cannot exclude that the cytoplasm contains low amounts of CRY1, the majority of translated CRY1 protein seems to enter the nucleus without any circadian gating. Similarly, we did not observe major cytoplasmic accumulation of PER2 protein prior to its nuclear appearance consistent with data from cells from the *Per2*-Venus mouse model[16]. Secondly, also in single cells, nuclear PER2 levels peak on average ~5 h before CRY1-levels, which is consistent with data from population sampling of murine cell nuclei[12,26,27]. This delay is also present on whole-cell protein level (Fig. 4c and Narumi[19]) indicating that in single cells circadian nuclear accumulation of CRY1 and PER2 mainly reflects the circadian expression levels of those proteins rather than being the consequence of circadian gating in nuclear appearance. Thirdly, quantification of fluorescence signal from fusion proteins indicated that nuclear CRY1 peak levels exceed those of PER2 by a factor of ~5–10 in U-2 OS cells. This is a much larger difference than that seen in mouse liver, where CRY1 peak expression level was reported to be only twice as high as that of PER2[19].

Together, these data raise the following questions: (i) Does CRY1 nuclear accumulation really directly depend on the presence of PER proteins as previously suggested[12]? PER2 levels seem not to be a limiting factor for CRY1 nuclear entry, since CRY1 is mainly nuclear regardless of PER2 expression phase (peak or trough). In addition, CRY1 levels peak when PER2 levels are already declining, and CRY1 is 5–10-fold more abundant than PER2. Biochemistry data indicate the existence of cytoplasmic complexes that contain CRY proteins but not PER2 in mouse liver[17], thus it is possible that other PER protein family members (PER1 and/or PER3) act as CRY1 carriers for nuclear entry. Our knock-in technology should now allow the efficient generation of PER1 and PER3 reporter cells to study this issue. It is also conceivable that a single PER2 protein may be able to shuttle multiple CRY proteins, e.g. in a sequential manner. The described dynamic shuttling of PER2-Venus protein in and out of the nucleus may support this[16]. An alternative explanation comes from our mathematical model of the cellular circadian oscillator: a shorter half-life time of nuclear PER2 compared to CRY1 could also explain the observed abundance differences. Indeed, while in a number of studies estimating PER2 and CRY1 half-life times the half-life values vary between studies, the trend is consistent with our modeling predictions, i.e. that CRY1 is more stable than PER2[16,23,67–70]. Probing both proteins' cellular stability using our reporter cells will help to clarify the underlying mechanism. (ii) Is most nuclear CRY1 protein present in a large negative feedback complex? Biochemistry data with murine liver lysates indicate that the majority of CRY1 protein indeed is present in a ~1.9-MDa negative feedback complex, which also includes CLOCK-BMAL1 and virtually all of the PER and CRY proteins as well as CK1δ[17]. Only a minority of nuclear CRY1 was found as monomer. Although the stoichiometry of clock proteins within the murine negative feedback complex has not yet been worked out, this seems to be different in U-2 OS cells. The delayed phase of nuclear CRY1 abundance compared to PER2, the fact that PER2 and CRY1 directly interact in a 1:1 ratio[9] and, more importantly, the much higher protein abundance at all circadian phases point

to a much higher degree of CRY1 proteins not being in a complex with PER2 in human cells. While CRY1 can act as PER-independent late repressor within the circadian clockwork[12,27–30], it will be interesting to study other PER-independent targets of CRY1, which may include nuclear receptors[71].

Our mathematical model simultaneously captured the three important experimental findings of our study (abundance, phase, and amplitude differences between PER2 and CRY1, Fig. 5b). Moreover, the model reproduced the positive amplitude-period correlation present in single-cell time series (Fig. 5c, d). Such dependencies are termed "twist" and have been discussed in the context of oscillator theory[58,72–74]. Theory predicts that "hard oscillators" lead to negative correlations between periods and amplitudes, while "soft oscillators" exhibit positive correlations. Thus, our data suggest that the circadian oscillators studied belong to the class of soft oscillators. The discovery of such dependencies, previously overlooked, might inspire new types of analyses. How generic are the twist dependencies? Are there "hidden" correlations of additional oscillation parameters? What is the role of the large CRY1-PER2 phase difference and is there, if any, an "optimal" phase angle for proper oscillator function? How is robustness of rhythms achieved at the single-cell level despite molecular noise and cell heterogeneity?

We are confident that single-cell time series data will contribute to the general paradigm for clock research, in which experimental data guide new modeling efforts, and modeling inspires new experimental hypotheses. This joint experimental/theoretical approach will likely lead to a deeper understanding of the fundamental properties that drive molecular circadian clocks in mammals. Using published algorithms that focus on loop reconstruction of oscillating systems[75], tools from stochastic modeling[76], or direct model fitting to the data (e.g. Almeida et al.[77]), future studies will aid in understanding the roles of PER2:CRY1 and CRY1 as repressors in the mammalian circadian clockwork, in a data-driven manner.

In summary, we present an efficient CRISPR-based knock-in strategy that allows the generation of reporter cells even for genes with low or variable expression levels, such as circadian clock genes. We created PER2 and CRY1 single and double knock-in reporter cells with clock proteins tagged to fluorescence proteins or luciferase. These enabled the visualization of PER2 and CRY1 protein dynamics in live single cells. Although individual cellular oscillators display a rather large degree of intercellular variability with respect to dynamic parameters, these new models propose several features of the (human) circadian oscillator that are not easily consistent with the canonical circadian oscillator model. Future studies are required to evaluate, whether these differences are due to the species, the tissue, the detection methods, or other unknown factors. We anticipate that the generation of additional single, double, or triple knock-in cells for circadian clock proteins will greatly advance our understanding about the cell biology of circadian clocks. Our study is the first step.

## Methods

**Cell lines.** U-2 OS (human, ATCC HTB-96) and HCT-116 (human, ATCC CCL-247) cells were cultured in DMEM supplemented with 10% FBS, 25 mM HEPES and penicillin/streptomycin at 37 °C and 5% $CO_2$. For long-term imaging, cells were cultured in FluoroBrite (GIBCO) medium supplemented with 2% FBS, 1x GlutaMax and penicillin/streptomycin from 2 days prior to imaging. The generated cell lines can be obtained from corresponding author.

**Plasmids.** The core sequence including positive selection cassette, LoxP sites, Frt-sites, His-Flag-tag, poly(A)-sites, and multiple cloning sites for insertion of homology arms was synthesized by commercial supplier (BaseClear) and cloned into pUC19 backbone. A negative selection cassette with human thymidine kinase was retrieved from Addgene #21911[78]. The hTK was exchanged for hCD4, amplified from the pMSCV-IRES-hCD4plasmid (Addgene #35712[79]). mClover3 was subcloned from Addgene #74252[38], mScarlet was subcloned from Addgene

#98839[80]. The pCAG-i53bp expression plasmid was a gift from Ralf Kuhn and is derived from Addgene #74939[45]. SV40-NLS CRE recombinase was a gift from Christoph Harms and was subcloned into pLenti6 backbone. CRY1 fusion proteins also included translation of the LoxP site C-terminal to the fluorophore to avoid nonsense-mediated decay. See Supplementary Table 1 for DNA sequence information.

Single guide RNAs (Supplementary Table 2) were designed to cut near the STOP codon using CRISPOR[81], and corresponding DNA oligos were ligated into pCRISPR-Lenti-v2 (Addgene #52961[82]) as described. To test efficiency of guides, cells were transduced with lentiviruses harboring the Cas9/sgRNA expression plasmid, puromycin resistant cells' gDNA was isolated, the respective region amplified by PCR and sequenced. Efficiency was assessed using TIDE assay[83].

pGIPZ clones V2LHS_172866 (CRY1) V2LHS_52938 (PER2) (Supplementary Table 3) were purchased from Open Biosystems (GE Healthcare) and the tGFP was mutated to abrogate fluorescence. The 0.9-kb Bmal1-promoter driven luciferase reporter construct is described in Maier et al.[84], and luciferase sequence for knock-in generation was subcloned from this plasmid.

**Transfection**. For knock-in experiments, $10^6$ cells were harvested by trypsinization and transfected with each 2 µg of i53bp, donor vector, and pCRSIPR-Lenti-V2 by electroporation using the NEON system (Thermo Fisher, buffer N, U-2 OS: 4 pulses, 10 ms, 1230 V, HCT-116: 2 pulses, 30 ms, 1130 V). For PER2-editing, a mixture of three sgRNA sequences was used in equimolar ratios. After electroporation, cells were seeded into antibiotic-free DMEM and cultured for 24 h before selection. Transient transfections of CRE recombinase were performed using 1 µL Lipofectamine 2000 and 200 ng CRE expression plasmid in a 48-well plate format.

**Virus production and transduction of cells using lentivirus**. HEK293-T cells were transiently transfected in a T75 flask with 8.6 µg lentiviral expression plasmid, 6 µg psPAX2, and 3.6 µg pMD2G (gift from the Trono lab, Addgene #12259 and #12260) packaging plasmid using CalPhos Mammalian Transfection Kit (Takara). Next day, culture medium was replaced by 12.5 mL complete culture medium, and lentiviral supernatant was collected after 24 and 48 h. Combined supernatant was passed through a 0.45-µm filter (Filtropur S 0.45) and either used directly or stored in aliquots at −80 °C. For transduction, cells were seeded into lentivirus containing supernatant supplemented with 8 µg/mL protamine sulfate. Next day, lentivirus containing supernatant was aspirated and cells were cultured in complete culture medium for further 24 h before antibiotic selection of transduced cells.

**Antibiotic selection**. To select for transfected or transduced cells, cells were grown sub-confluently in blasticidin (10 µg/ml) containing medium for >3 days or in puromycin (10 µg/ml) containing medium for >1 day, until non-transfected control cells died.

**FACS sorting**. Cells were sorted on a FACS AriaII (BD). For staining of surface hCD4 for negative selection, $2 \times 10^6$ cells were trypsinized, washed with 0.5% BSA/PBS and incubated with 200 µL of a 1:50 dilution of hCD4-BV711 (OKT4, Bio-Legend, UK) for 30 min. Cells were washed twice with BSA/PBS. Excitation: 405 nm. Emission filter: 525LP-525/50 (CFP), 685LP-710/50 (BV711). The sorting strategy is shown exemplarily in Supplementary Fig. 10.

**Nucleic acid isolation and PCR**. Genomic DNA was extracted using direct PCR Lysis Reagent Cell (VWR) for PCR analysis or using Qiagen Blood and Tissue DNA isolation kit for WES and copy number analysis. RNA isolation was performed using the AMBION PureLink RNA Mini kit (Themo Fisher) according to the manufacturer's instruction, including an on-column DNase digest. RNA was reversely transcribed using gene-specific primer and a two-step protocol. PCR amplifications were performed using Phusion polymerase (New England Biolabs), products were analyzed by agarose gel electrophoresis and detected using RedSafe/UV light. Primer sequences are listed in Supplementary Table 4. Of note, additional low mobility bands were observed from PCR of one-allelic knock-in clones (Supplementary Figs. 1h and 6c). Sanger sequencing of one of those bands revealed a mixture of wild-type and knock-in sequence. This and the fact that we did not observe low mobility bands in PCRs from bi-allelic knock-in clones made us conclude that they represent slowly migrating heteroduplexes and not additional PCR products.

**Whole exome sequencing and off-target analysis**. WES was performed by Novogene (https://en.novogene.com), using the Agilent SureSelect Human All ExonV6 kit and 1.0 µg genomic DNA per sample for sequencing library generation. The raw NGS data was filtered for adapter contamination, overrepresentation of 'N's and low quality. The Next Generation Sequencing reads were aligned by BWA-MEM (0.7.17-r1188)[85]. Potential Cas9 off-target sites were predicted using four different tools (CasOFFinder[86], allowing for up to four mismatches and RNA/DNA bulge of 1, CRISPR-ML[87,88], CCtop[89], and CRISPOR[81]). WES data were aligned to the reference genome by Novogene to hg38 and indels were detected using GATK(v4.0). Indels present in the founding wild-type population or in U-2 OS wild-type clones that we had analyzed previously[90] were filtered out. We then examined, whether potential Cas9 off-target cut sites overlap with or are in close

vicinity (±20 bp) to unique indels. To examine potential random integration of the donor vector to exonic sites, the NGS reads were re-aligned using a reference constructed from the primary chromosomes assembly for GRCh38 and the donor vectors. Finally, the alignments were searched for the presence of reads or read-pairs aligning to both the donor plasmid and exonic regions. For more details, see Supplementary Note 1.

**Copy number analysis by digital droplet PCR**. Primers and probes for ddPCR were designed using Primer3 software[91]. Primer specificity was tested using standard PCR on knock-in clones to yield only unique products. Sanger sequencing confirmed product identity. ddPCR was performed on a Biorad QX200 system according to the manufacturer's recommendations. ddPCR Supermix without dUTP was used for amplification. Dual-labeled probes were synthesized by Microsynth (Germany) and contained either FAM or HEX at the 5′-end and BHQ1 quencher at the 3′-end. Annealing/elongation temperature was set to 60.8 °C. Five units of BstY1 restriction enzyme (NEB) per 20 µl reaction was added directly to the reaction mix. Sequences of primers and probes as well as final probe concentrations are listed in the Supplementary Tables 1 and 5. Data were analyzed by QuantSoft software (www.bio-rad.com).

**Bioluminescence recording of circadian oscillation**. Luciferase knock-in cells or fluorescent knock-in cells transduced with an mBmal1 promoter driven luciferase reporter plasmid were seeded to reach confluence. To synchronize the circadian rhythms, cells were either treated with 1 µM dexamethasone for 30 min followed by washing with warm PBS (Fig. 4c and Supplementary Fig. 8), or were washed twice with cold PBS for 2 min (Supplementary Figs. 4 and 6). Cells were then incubated in DMEM without phenol-red supplemented with 250 µM D-luciferin, and dishes were sealed using parafilm. Bioluminescence was recorded in a 96-well plate luminometer (TopCount, Perkin Elmer) or in a LumiCycle (Actimetrics). Raw data were detrended by dividing by the 24-h running average. Periods, phases and mean bioluminescence signal were estimated by fitting the cosine wave function using the ChronoStar software[92].

**Microscopy**. For microscopy, cells were seeded on glass bottom #1.5H µ-slides (IBIDI, Germany) or glass bottom #1.5H-N 96-well plates (Cellvis, USA). Imaging was performed on a Nikon Widefield Ti2 equipped with a sCMOS, PCO.edge camera and a live-cell incubator. Image acquisition was done in Flurobrite medium (GIBCO) supplemented with 2% FBS, 1:100 PenStrep, and 1x GlutaMax at 37 °C and 5% $CO_2$. The following light sources (LEDs) and emission filters were used for the different channels: CFP (cerulean): excitation 438/29, emission 473/24 nm; GFP (dClover2): excitation 475/28 nm, emission 520/26 nm; YFP (mClover3, dClover2): excitation 511/16 nm, emission 540/30 nm; RFP (mScarlet-I) excitation 555/28 nm, emission 642/80 nm. Objectives: 40x ApoFluor, NA 0.95, WD 250 µm; 20x Plan Apo, NA 0.8, WD 1 mm. Illumination time for CFP was usually 500 ms and 2 s for all other channels. To synchronize the circadian rhythms, cells were either washed twice with cold PBS for 2 min (Fig. 1f, Supplementary Figs. 3 and 7a) or treated with 1 µM dexamethasone for 30 min followed by washing with warm PBS (Figs. 2–4), or by exchange to pre-warmed imaging medium (Supplementary Figs. 5 and 7b, c). Imaging started 2 h after synchronization with a regular imaging interval of 1 h.

**Fluorescence data analysis**. Fluorescence data were extracted using ImageJ. Nuclei or cytoplasm were manually marked using either the respective fluorescence channel (CRY1 positive nuclei) or phase contrast (all other nuclei and cytoplasm), and mean fluorescence intensities were extracted. For Figs. 2–4, individual background fluorescence for every cell at each time point was determined by quantifying the same area of the imaging field from a cell-free image frame of the same experiment. Mean background signal was subtracted. Linear trends were detected by linear regression analysis of all cells from one imaging frame and also eliminated by subtraction. Since trough PER2-fluorescence levels were not surpassing background noise, we set the lowest intensity of each time series to 0. For Supplementary Figs. 5 and 7c, mean background signal was averaged from four cell free areas per time-point and subtracted. For Supplementary Figs. 3f, 4f, g, and 7a, b, matching background ROIs were defined for every cell at each time point, and mean background was subtracted. To eliminate cell division outliers, fluorescence values at cell division events were imputed by averaging across neighboring time points.

The data was exported, processed and analyzed in R (version 3.6). Rhythmicity of single-cell time-series was evaluated with meta2d from the MetaCycle R package (version 1.2[50]), with minper=20, maxper=28, and cycMethod=c ("ARS", "JTK", "LS"), thus incorporating the ARSER, JTK_CYCLE and Lomb-Scargle algorithms[93–95]. For data in Supplementary Fig. 7b, only Lomb-Scargle algorithm was used due to the nature of the raw data. Time series that did not pass the rhythmicity test (Benjamini-Hochberg FDR > 0.05) were excluded from further analysis. Data were normalized for Figs. 2e and 3d by dividing intensities by mean intensities of the respective time series. Plots were drawn using ggplot2 v3.2. To facilitate comparison of phases, extracted phase was divided by extracted period (mean period for double knock-in cells) and multiplied by 24. When comparing phases of double knock-in cells, we chose the direction (advance or delay) according to which absolute resulting phase difference was lower than 8, or, when in rare cases above 8, closer to the mean of all other data.

**Semi-quantitative analysis of fluorescence signals**. To confidently compare fluorescence data for the different fusion proteins with the same fluorophore entity, we made the following assumptions and analyzed and normalized the data accordingly: (i) Because PER2 trough expression levels were not distinguishable from background, we compared peak expression levels that can be quantified with higher confidence. (ii) As we assumed that the distribution of peak protein expression is similar for different clones, we extracted peak fluorescence values from each background-subtracted time series in a time window of either the first or the second day of recording and compared the means. (iii) Since lowest PER2 intensity were set to 0 (see above), we added the average nuclear background signal intensity (Supplementary Fig. 4g) to all PER2 peak values (34.0 a.u.). Conceptually, this constant can be regarded as the lower limit of mScarlet-I detection. This correction reduces the underestimation of fusion protein expression from low signals. All data were normalized to the mean of PER2-mScarlet-I peak expression (Fig. 4d).

To be able to estimate protein ratios in single cells, in which PER2 and CRY1 are fused to different fluorophores, we first had to determine the relative signal intensities of mScarlet-I and mClover3 resulting from a similar amount of fusion proteins in our experiment. We compared mean peak intensities (1st and 2nd peak independently) of CRY1-mClover3 (divided by two because of the bi-allelic knock-in) with those of CRY1-mScarlet-I from single-knock-in or double knock-in cells. We estimated that in average, mScarlet-I intensities surpass mClover3 intensities resulting from a supposedly similar amount of fluorophores by a factor of 4.3 under the used experimental settings. By correcting for this, and for the detection limit of PER2 (see above), the ratio of CRY1-mClover to PER2-mScarlet was determined using Eq. (1):

$$\text{ratio}\left[\frac{\text{CRY1}}{\text{PER2}}\right] = \frac{\text{fluorescence(CRY1-}m\text{Clover3} - \text{BG)} \times 4.3}{\text{fluorescence(PER2-}m\text{Scarlet-}I - \text{BG)} + 34} \quad (1)$$

**Fluorescence correlation spectroscopy**. FCS measurements were performed using an Olympus FluoView FV-1000 system (Olympus, Tokyo, Japan) equipped with a time-resolved LSM upgrade (PicoQuant GmbH, Berlin, Germany) with a ×60 UPlanSApo NA 1.2 water immersion objective. For FCS measurements, a 483 nm or a 561 nm pulsed diode laser (Picoquant GmbH, Germany) was used, emission light was passed through a 540/20 nm (mClover3) or 595/50 nm (mScarlet-I) and detected via SPAD detector.

FCS data were acquired and analyzed using SymphoTime 64 software (PicoQuant GmbH, Berlin, Germany). The confocal volume was calibrated using Atto-488 dye with a diffusion constant of 400 μm²/s. Background level and average background were estimated by measurement of 15–30 corresponding nuclear areas of non-fluorescent cells for each channel. For each cell, five nuclear measurement areas were chosen on the basis of a differential interference contrast (DIC) image and measured for 10 s in each channel (pixel dwell time of 100 μs, >5 μW laser power). Auto-correlation curves were calculated using automated fluorescence lifetime background correction, and curves were fitted to a simple diffusion model. For measurements with fluorescence not exceeding background levels or when the fit did not result in concentration values, fluorophore concentration was set to an estimated detection limit (10th percentile of all measurements above background, 0.11 nM). Results with counts-per-molecules below a threshold (mean − 3 × standard deviation of quartile range) were not considered. Background correction was performed as suggested by Schwille and coworkers[96] using eq. (2):

$$c_{\text{corrected}} = c_{\text{measured}} \times \frac{(F_m - F_{\text{BG}})^2}{F_m{}^2} \quad (2)$$

with $c$ being the concentration, $F_m$ the measured fluorescence intensity, and $F_{\text{BG}}$ the average background fluorescence intensity in counts per second. The result was divided by 0.6 to correct for the estimated proportion of non-fluorescent states of the fluorophores[97] to obtain the final concentration.

**Fluorescence lifetime imaging**. Fluorescence lifetime imaging (FLIM) experiments of cells expressing fusion proteins were performed using a specialized laser-scanning microscope based on a commercial scan head (TriMScope II, LaVision BioTec, Bielefeld, Germany). A near-infrared laser (Ti:Sa, Chameleon Ultra II, Coherent, Dieburg, Germany) tuned at 930 nm, repetition rate 80 MHz was used as excitation source for mClover3. A water-immersion objective lens (×20, NA 1.05, Apochromat, Olympus, Hamburg, Germany) was used to focus the laser beam into the sample. mClover3 fluorescence and auto-fluorescence were collected in the backward direction using a dichroic mirror (775, Chroma, US), passed through an interference filter (525 ± 25 nm) and was detected by a GaAsP PMT connected to time correlated single photon counting (TCSPC) electronics (LaVision BioTec – a Milteny company, Bielefeld, Germany). The TCSPC data, i.e. photon counting histograms in each pixel of the image, were collected at a time resolution of 55 ps, over 12.48 ns. In all imaging experiments, we used an average maximum laser power of 10 mW to avoid photo-toxicity. Each image had a field-of-view of 100 μm × 100 μm and a digital resolution of 248 × 248 pixel. The fluorescence decays in each image were evaluated using the phasor approach to FLIM[98]. The result of this evaluation is the image of the mean fluorescence lifetime of all detected fluorescence signals in each pixel as shown in Supplementary Fig. 1f. Reported fluorescence lifetimes are means ± SD of 7 cells, each with ~2000 pixels evaluated for nuclear and ~1000 pixel for perinuclear signals.

**Mathematical modeling**. We modified an already published[53] model of the mammalian clockwork and translated our experimental findings to refine the system. Numerical simulations were performed in Python using the multiflap Python toolbox to solve non-linear systems of ordinary differential equations[99]. Cell-to-cell heterogeneity was simulated by randomly and simultaneously varying all transcription, translation, degradation as well as nuclear import and export rates. For more details on model generation, please refer to Supplementary Note 2 and Fig. 5a.

**Statistics and reproducibility**. Normality of data sets was tested using D'Agostino-Person normality test. For normally distributed datasets student's t-test or one-way ANOVA were used to calculate p-values. One-way ANOVA was also used for the data in Supplementary Fig. 4f, g, where normality test was not possible due to small sample size. For non-normally distributed data, Mann–Whitney-U and Kruskal–Wallis test were used to calculate p-values for unpaired data, and Wilcoxon matched-pairs signed rank test for paired data. Rhythmicity was determined using MetaCycle as described in data analysis. All p-values are from two-tailed tests. Test statistics are listed in Supplementary Tab. 8.

Figure 1c: transfection experiments with the described constructs were performed once as they yielded the desired clones. The PCR experiments were repeated three times with similar results. Figure 1d: Cre transfection was performed once for each transfected cell population (eight times in total for the cell lines presented) and typically yielded 15–60% CFP negative cells. Figure 1e: images are representative for >3 independent examinations of the same clone. The number of clones showing the same pattern are shown in Supplementary Fig. 1j. Figure 1f: each image shown is representative of five imaging areas. The knockdown experiment was performed once. Figure 2a–d: time series montages are examples from (top to bottom) 19, 22, 20, 20 tracked and analyzed cells whose extracted data is shown in Fig. 2e. Similar oscillatory patterns over the course of 3 days were observed in two other independent experiments. Figure 3a: images are representative for >3 independent examinations of the same clone. The number of clones showing the same pattern is shown in Supplementary Fig. 6. Figure 3b, c: time series montages are examples from 50 tracked and analyzed cells whose extracted data are shown in Fig. 3d. Similar oscillatory patterns over the course of three days were observed in two other independent experiments. Supplementary Figure 1a, b: for PCR-confirmed knock-in clones, images are representative for >3 other independent examinations of the same clones. Clones without desired knock-in were not re-imaged. The number of clones that showed the same pattern is shown in Supplementary Fig. 1j. Supplementary Figure 1c–e: images are representative of >3 independent evaluations of the same clones. Supplementary Figure 1f: image is representative of ~20 cell in five imaging regions aquired in a single experiment. Supplementary Figure 1g, h: PCR experiments were repeated twice with similar results. Supplementary Figure 3b–e: each image shown is representative of five imaging regions. The knockdown experiment was performed once. Supplementary Figure 5a: time series montages are examples from 10 tracked and analyzed cells of a clone from a single experiment. Similar oscillatory patterns were observed for a second clone. Supplementary Figure 6b, c: PCR experiments were repeated twice with similar results.

**Reporting summary**. Further information on experimental design is available in the Nature Research Reporting Summary linked to this paper.

## Data availability

Source data are provided with this paper. WES data are available in the NCBI Sequence Read Archive, https://www.ncbi.nlm.nih.gov/sra/PRJNA677859. Raw imaging data associated to Figs. 1f, 2, 3, 4a, b, d, e, 5c, and Supplementary Fig. 3 are available at the Biostudies archive (https://www.ebi.ac.uk/biostudies/studies/S-BSST630 and https://www.ebi.ac.uk/biostudies/studies/S-BSST631). All other datasets generated and/or analyzed as part of this study are available from the corresponding author upon reasonable request. Source data are provided with this paper.

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

# ARTICLE

67. Hirano, A., Braas, D., Fu, Y.-H. & Ptáček, L. J. FAD regulates CRYPTOCHROME protein stability and circadian clock in mice. *Cell Rep.* **19**, 255–266 (2017).

68. Hirano, A. et al. In vivo role of phosphorylation of cryptochrome 2 in the mouse circadian clock. *Mol. Cell. Biol.* **34**, 4464–4473 (2014).

69. Siepka, S. M. et al. Circadian mutant overtime reveals F-box protein FBXL3 regulation of cryptochrome and period gene expression. *Cell* **129**, 1011–1023 (2007).

70. Jang, J. et al. The cryptochrome inhibitor KS15 enhances E-box-mediated transcription by disrupting the feedback action of a circadian transcription-repressor complex. *Life Sci.* **200**, 49–55 (2018).

71. Kriebs, A. et al. Circadian repressors CRY1 and CRY2 broadly interact with nuclear receptors and modulate transcriptional activity. *Proc. Natl. Acad. Sci. USA* https://doi.org/10.1073/pnas.1704955114 (2017).

72. Ashwin, P., Coombes, S. & Nicks, R. Mathematical frameworks for oscillatory network dynamics in neuroscience. *J. Math. Neurosci.* **6**, 2 (2016).

73. Winfree, A. *NonlinearDynamics: Where Do We Go From Here?* (eds. Hogan, J. et al.) 229–268 (Taylor & Francis, 2002).

74. Myung, J. et al. The choroid plexus is an important circadian clock component. *Nat. Commun.* **9**, 1062 (2018).

75. Pigolotti, S., Krishna, S. & Jensen, M. H. Oscillation patterns in negative feedback loops. *Proc. Natl Acad. Sci. USA* **104**, 6533–6537 (2007).

76. Andrews, S. S., Dinh, T. & Arkin, A. P. *Encyclopedia of Complexity and Systems Science*. 8730–8749 (Springer New York, 2009).

77. Almeida, S., Chaves, M. & Delaunay, F. Transcription-based circadian mechanism controls the duration of molecular clock states in response to signaling inputs. *J. Theor. Biol.* **484**, 110015 (2020).

78. Dewey, R. A. et al. Chronic brain inflammation and persistent herpes simplex virus 1 thymidine kinase expression in survivors of syngeneic glioma treated by adenovirus-mediated gene therapy: implications for clinical trials. *Nat. Med.* **5**, 1256–1263 (1999).

79. Artinger, E. L. et al. An MLL-dependent network sustains hematopoiesis. *Proc. Natl. Acad. Sci. USA.* **110**, 12000–12005 (2013).

80. Mastop, M. et al. Characterization of a spectrally diverse set of fluorescent proteins as FRET acceptors for mTurquoise2. *Sci. Rep.* **7**, 11999 (2017).

81. Haeussler, M. et al. Evaluation of off-target and on-target scoring algorithms and integration into the guide RNA selection tool CRISPOR. *Genome Biol.* **17**, 148 (2016).

82. Sanjana, N. E., Shalem, O. & Zhang, F. Improved vectors and genome-wide libraries for CRISPR screening. *Nat. Methods* **11**, 783–784 (2014).

83. Brinkman, E. K., Chen, T., Amendola, M. & van Steensel, B. Easy quantitative assessment of genome editing by sequence trace decomposition. *Nucleic Acids Res.* **42**, e168–e168 (2014).

84. Maier, B. et al. A large-scale functional RNAi screen reveals a role for CK2 in the mammalian circadian clock. *Genes Dev.* **23**, 708–718 (2009).

85. Li, H. Aligning sequence reads, clone sequences and assembly contigs with BWA-MEM. *arXiv* https://arxiv.org/abs/1303.3997 (2013).

86. Bae, S., Park, J. & Kim, J.-S. Cas-OFFinder: a fast and versatile algorithm that searches for potential off-target sites of Cas9 RNA-guided endonucleases. *Bioinformatics* **30**, 1473–1475 (2014).

87. Listgarten, J. et al. Prediction of off-target activities for the end-to-end design of CRISPR guide RNAs. *Nat. Biomed. Eng.* **2**, 38–47 (2018).

88. Doench, J. G. et al. Optimized sgRNA design to maximize activity and minimize off-target effects of CRISPR-Cas9. *Nat. Biotechnol.* **34**, 184–191 (2016).

89. Stemmer, M., Thumberger, T., del Sol Keyer, M., Wittbrodt, J. & Mateo, J. L. CCTop: an intuitive, flexible and reliable CRISPR/Cas9 target prediction tool. *PLoS ONE* **10**, e0124633 (2015).

90. Nikhil, K. L., Korge, S. & Kramer, A. Heritable gene expression variability and stochasticity govern clonal heterogeneity in circadian period. *PLoS Biol.* **18**, e3000792 (2020).

91. Untergasser, A. et al. Primer3—new capabilities and interfaces. *Nucleic Acids Res.* **40**, e115–e115 (2012).

92. Maier, B., Lorenzen, S., Finger, A.-M., Herzel, H.-P. & Kramer, A. *Searching novel clock genes using RNAi-based screening: Circadian Clocks: Methods and Protocols* (ed. Brown, S.) (Springer, 2020).

93. Hughes, M. E., Hogenesch, J. B. & Kornacker, K. JTK_CYCLE: an efficient nonparametric algorithm for detecting rhythmic components in genome-scale data sets. *J. Biol. Rhythms* **25**, 372–380 (2010).

94. Glynn, E. F., Chen, J. & Mushegian, A. R. Detecting periodic patterns in unevenly spaced gene expression time series using Lomb-Scargle periodograms. *Bioinformatics* **22**, 310–316 (2006).

95. Yang, R. & Su, Z. Analyzing circadian expression data by harmonic regression based on autoregressive spectral estimation. *Bioinformatics* **26**, i168–i174 (2010).

96. Schwille, P., Haupts, U., Maiti, S. & Webb, W. W. Molecular dynamics in living cells observed by fluorescence correlation spectroscopy with one- and two-photon excitation. *Biophys. J.* **77**, 2251–2265 (1999).

97. Dunsing, V. et al. Optimal fluorescent protein tags for quantifying protein oligomerization in living cells. *Sci. Rep.* **8**, 10634 (2018).

98. Leben, R., Köhler, M., Radbruch, H., Hauser, A. E. & Niesner, R. A. Systematic enzyme mapping of cellular metabolism by phasor-analyzed label-free NAD (P)H fluorescence lifetime imaging. *Int. J. Mol. Sci.* **20**, 5565 (2019).

99. Ducci, G., Colognesi, V., Vitucci, G., Chatelain, P. & Ronsse, R. Stability and sensitivity analysis of bird flapping flight. *J. Nonlinear Sci.* **31**, 47 (2021).

## Acknowledgements

We thank all current and former members of the Kramer and Herzel labs for technical and intellectual assistance. We acknowledge the assistance of the FCCF at the Deutsches Rheuma-Forschungszentrum. Furthermore, we thank the Advanced Medical Bioimaging Core Facility (AMBIO) of the Charité for support in acquisition of the imaging data. We thank Mir-Farzin Mashreghi, Joachim Fuchs, Christoph Harms, Steven Brown and Michela Di Virgilio for providing material, Annika Nonnenmacher and Janina Gabriel for assistance on data acquisition and analysis, and Valentin Dunsing, Bharath Ananthasubramaniam, Christoph Schmal and Gianmarco Ducci for helpful discussions. Adrián E. Granada's research was supported by the German Federal Ministry for Education and Research (BMBF) through the Junior Network in Systems Medicine, under the auspices of the e:Med Programme (grant 01ZX1917C). This work was funded by the Deutsche Forschungsgemeinschaft (DFG, German Research Foundation) - Project Number 278001972 - TRR 186.

## Author contributions

A.K. conceived and supervised the project. C.H.G. and S.R. designed the knock-in strategy. C.H.G., H.D., B.K. and A.G. generated the cell lines and acquired data. T.K. and A.H. contributed to F.C.S. data acquisition and analysis. M.O. and H.H. designed and analyzed the mathematical model. C.U., A.R. and R.N. acquired and analyzed FLIM data. M.J. and T.Z. analyzed and interpreted NGS data. B.M. and A.Z. contributed to raw data processing and interpretation. C.H.G., M.O. and A.K. wrote the manuscript and prepared the figures. H.H., H.E. and A.E.G. revised the manuscript and gave important methodical advice.

## Funding

## Competing interests

The authors declare no competing interests.
