## [Peer Review File · Nature Communications]

Reviewer comments, first round –

Reviewer #1 (Remarks to the Author):

This is an interesting study that integrates the development of CRISPR-mediated methods to tag low abundance and/or transiently expressed genes with some cell biology insight into mechanisms of circadian rhythms using the fluorescently-tagged clonal lines they developed. They generated knock-ins of fluorescent proteins (mClover3 or mScarlet-I) at two clock genes, *Per2* and *Cry1*, using a clever selection strategy that offers several advantages for targeting clock genes, which are typically expressed transiently at specific times of the day at low levels. Importantly, their methods allow removal of the positive selection cassette inserted downstream of the gene by transfecting with a Cre expression plasmid, restoring endogenous 3'-UTRs, which are frequently known to play a role in post-transcriptional regulation of gene expression. These methodological developments should be of broad interest to those studying non-abundant genes.

After thorough screening for both singly or doubly-labeled cells (e.g., cells with *CRY1*-mClover3 and *PER2*-mScarlet-I), they studied cellular circadian dynamics of the two clock proteins, demonstrating circadian changes in their abundance and nuclear localization that were largely expected based on prior data. The omission of key citations that previously established the delayed phase of *Cry1* expression and the importance of this in circadian rhythms detracts a bit from the conclusions of this study by not putting them in the context of what is already known in the field. However, the unique tools developed here allowed for the observation that *CRY1* protein abundance is likely in ~6-8-fold excess relative to *PER2* in intact cells, calling into question an over-simplified canonical model of circadian repression that all of the *CRY* and *PER* proteins are bound in large complexes with one another. The cellular tools and careful quantification of *CRY1* and *PER2* levels in cycling cells here contribute to an emerging model where the delayed expression of *CRY1* gives rise to its unique role in the mammalian circadian clock (see below for details). With the inclusion of several citations to properly attribute earlier observations of delayed *CRY1* expression, nuclear accumulation, and function in the clock, these data provide new evidence of, and new tools to explore, the delayed role of *CRY1* in the clock. The questions raised here about the dependence of *CRY1* on *PER2* for nuclear entry and its activity outside of (and after) the large *PER*-*CRY* complexes that dominate the early part of repression should be of interest to the circadian community. Overall, there are a few concerns and missed opportunities to put these data in the context of prior data and models, but these should be easily addressable.

Major:

1. The data in Figure 2I (and elsewhere) provide strong evidence for a significant delay in the peak of *CRY1* entry into the nucleus; however, there is no citation of the publication that first discovered the phase-delayed expression of *Cry1* (Ukai-Tadenuma et al. (2011) *Cell*) or several that followed, demonstrating that the delayed expression encoded by the *Cry1* promoter is essential for reconstitution of circadian clocks in cells and in vivo (Khan, S. et al. (2012) *J Biol Chem*, Li, Y. et al. (2016) *BBRC*, Rosensweig, C. et al. (2018) *Nat Comm*, and Maywood, E. et al. (2018) *PNAS*).
2. The language surrounding 'the current model of the oscillator' seems a bit oversimplified. We agree that language surrounding the prevailing model tends to be a bit dogmatic, but there are experimental data from mice supporting a unique, *PER*-independent role for *CRY1* late in repression going back as far as Stratmann, M. et al. (2010) *Genes & Dev* and Koike, N. et al. (2012) *Science*, and explicit models have been put forth since Ye, R. et al. (2014) *Genes & Dev*, Gustafson, C. and Partch, C. (2014) *J Mol Biol*, and Merbitz-Zahradnik, T. and Wolf, E. (2015) *FEBS Letts*. Presenting the current state of knowledge as though there is just one model, especially in the face of your data, seems like it incompletely addresses the state of the field.
3. The text in the Discussion that "In mouse liver, biochemistry data suggest that *CRY* and *PER* proteins almost exclusively coexist in huge cytosolic and nuclear complexes suggesting common regulation" also does not capture the full breadth of literature on this topic. First, we encourage

the authors to look at western blots in Supp. Fig. S1 for the paper they cite here (Aryal, R. et al. (2017) Mol Cell), which indeed show a considerable amount of CRY1 that is not sequestered in large PER-CRY complexes, particularly at later phases in the cycle. Second, this statement also neglects to cite biochemical ChIP-Seq data from mouse livers from the Takahashi lab (Koike, N. et al. (2012) Science) that also shows the apparent PER-independent recruitment of CRY1 to CLOCK:BMAL1 late in the repressive cycle. Presenting a more holistic view of the field would strengthen the clock-specific contributions of this manuscript.

Minor:

1. In supplementary figure 1G, the bottom panel was not labeled like the upper panel (i.e., PCR: PER2) and it is not clear where the arrows are pointing to indicate positive clones--is the PER2-FP arrow pointing to the ladder of bands in the middle lane as the positive clone? Perhaps something is off in the alignment of the arrowheads?

2. It is stated in Supplementary figure 4A-D that the data represent 4 independent runs from 2 experiments, but $n = 2$ points are shown for period and amplitude and lack statistical analyses. Were statistical tests run on the full dataset to support the claim that there is no difference between the clonal lines and wild-type cells?

3. Although there is valuable information in tracking CRY1 and PER2 fluorescence immediately after synchronization by dexamethasone (e.g., Fig. 2A-D but particularly PER2 expression in 2D), it is common practice to omit or gray out data from the first 12-24 hours to focus on expression driven later by circadian oscillations. Could you annotate this first peak or remark on this somewhere for those who are not in the circadian field?

4. The PER2-mScarlet-I cells show robust circadian expression of the tagged protein, which may be due to its overall higher fluorescence compared to the PER2-mClover3 clone (~10X lower). However, this is not readily visible in either the PER2-mClover3 or CRY1-mScarlet-I HCT-116 cells shown in Supp. Fig. 5. Is the punctate staining here also due to auto-fluorescence? These data do not seem to provide strong evidence that recapitulates the U-2 OS data.

5. While it is expected that the intensity of the first peak of fluorescence in doubly tagged cells (CRY1-mClover3 and PER2-mScarlet-I) would be the highest in Fig. 3C, it is curious (and interesting) that the second is of quite low fluorescence intensity, while the 3rd is much higher. This is not commonly observed with clock protein-luciferase fusions (i.e., PER2::LUC and Fig. 4 here)--has this been observed with any of the prior studies fluorescently tagging clock proteins? Is there any explanation for this?

6. The suggestion in the Discussion that "it is also conceivable that a single PER2 protein may be able to shuttle multiple CRY proteins, either at a time or in a sequential manner" does not seem to be plausible given that each PER protein has a single well-studied and high-affinity CRY-binding site. Are you suggesting that there is another cryptic CRY-binding site on PER2, or that dynamic nuclear shuttling of PER2-Venus is somehow correlated with the release of CRY proteins in the nucleus for the sequential transport of multiple CRY proteins? While this could be true, the data here fall far short of probing anything like this and it seems like a bit of a reach, even for a discussion.

Reviewer: Carrie Partch, UC Santa Cruz

Reviewer #2 (Remarks to the Author):

In accordance with the Editor's request, this reviewer provides a review with aspects related to CRISPR gene editing. I leave the evaluation of other aspects of the manuscript to other referees.

In the context of genome editing, the authors created knock-in cells carrying fluorescence reporter genes at the CRY1 and PER2 loci. The authors' method includes a floxed positive selection cassette and a negative selection cassette at the outside of the homology arm. This approach should be particularly useful for tagging the endogenous genes with low expression.

The authors describe their method as a "new strategy"; however, there is no technologically novel point. The insertion of fluorescence reporter gene nearby the stop codon is a common maneuver

for the visualization of endogenous gene expression. Floxed positive selection cassette and the negative selection to avoid the random integration are also the well-characterized strategies. Coexpression of i53bp was previously reported, too. The knock-in strategy itself is also a standard HDR-mediated gene insertion. This reviewer thinks this report does not contain sufficient technological advance to consider publishing in Nature Communications unless there are sufficient novelty and impact in other aspects such as the dynamics analysis of clock proteins.

Additional specific comments:

1. Off-target analysis should be fully performed. Treatment of i53bp and co-introduction of three sgRNAs may increase the potential risk.
2. Genotyping experiments are not sufficient. The authors only performed standard PCR and sequencing. Southern blotting and/or copy number analysis using ddPCR quantification should be additionally performed to demonstrate the non-existence of random integration and the proper on-target genotype.
3. Regarding the last sentence of the Abstract, triple knock-in was not supported in this study at all.

Reviewer #3 (Remarks to the Author):

This study by Gabriel et al on "Live-cell imaging of circadian clock protein dynamics in CRISPR-generated knock-in cells" represents one of the earliest attempts to generate combinations of fluorescent clock protein reporters at their endogenous loci in the same cells using CRISPR technique. This approach is intrinsically challenging because 1) the potential "off-target" nature of CAS9 by random integration of the donor sequence into the genome; 2) the low abundance (hence low signal) of clock transcription factors. The authors have taken great caution in generating and characterising CRISPR knock-in cell clones, with clever tricks (including both positive and negative selection process) to minimise the two problems mentioned above. They have targeted two highly rhythmic clock factors CRY1 and PER2 and tagged them with various FP and LUC tags. To make the results directly comparable, they also generated double knock-in cells with PER and CRY tagged by different FPs.

These efforts are paid off in that several new insights have been obtained including the visualisation of both PER2 and CRY1 in the same cells, with the most exciting discovery being the nuclear localisation of CRY1 throughout the circadian cycle without clear circadian gating (contrary to current models based on low resolution biochemical analysis). In addition, CRY1 is phase delayed by ~5 hours as compared to PER2 in the same cells (based on both FP and LUC reporters). Although PER2 has been extensively studied in fruit flies and mice (including PER2::VENUS and PER2::LUC), there have been no critical studies of tagging endogenously expressed CRY1 proteins using real-time reporters in cells, and nor studies to tag both PER and CRY in the same cells. As such, this paper represents both technological and biological advances in our understanding of circadian cell biology.

The experiments are overall very well designed with proper controls and validations in different cell lines (U20S, HCT), different synchronisation methods (dex vs temperature) and different types of reporters (FP vs LUC). Functional tagged clock proteins were confirmed by sequencing of junctions, shRNAs and by sustained circadian rhythms in cells. There are a few weaknesses that could be improved by in depth discussion, analysis or additional experiments:

- 1) It seems a missed opportunity that the authors did not attempt to build a mathematical model based on the semi-quantitative imaging data (phase/amplitude/abundance difference between PER and CRY), despite being experts in the clock modelling field.
- 2) All the PER2-FP reporters showed rather weak signals, limiting their utility. This is partly due to the low abundance of PERs. Is it also possible that the large PER proteins somehow interfere with the folding or maturation time of FP tags? At the trough level, it is very hard to tell whether there is any real signal above background. The authors stated: "however, a reliable discrimination from

auto-fluorescence and background signals was not possible". Have the authors considered using spectral lambda imaging and linear unmixing to eliminate auto-fluorescence?

3) One of the main conclusions is the 6-10 fold difference in abundance between PER2 and CRY1, based on fluorescence intensity. Given that PER and CRY are very different proteins (molecular weights, PTMs, structures), the FP signal intensity is likely to be skewed by such intrinsic differences (even when the same FP was tagged on different proteins). Therefore, this conclusion needs to be validated by additional quantitative imaging (for example, measuring # of molecules in a confocal volume) or biochemical methods (Western Blot taking advantage of the His flag tag in the constructs).

4) A key part of the result section has been dedicated to the generation and validation of the CRISPR cell lines. Even though these methodologies are important for peers in future efforts, it somehow dilutes the key biological insights. I wondered whether some of these could be moved to Methods section?

5) U2OS is an osteosarcoma-derived cell line. It would be interesting to discuss whether the conclusions drawn in U2OS would be broadly applicable to primary normal cells and the SCN.

6) In discussion, "This delay is also present on whole-cell protein level (Fig. 4C and 28) indicating that in single cells circadian nuclear accumulation of CRY1 and PER2 mainly reflects the circadian expression of those proteins rather than being the consequence of circadian gating in nuclear appearance.". I think degradation of these repressors should play a role too, not just circadian expression.

Reviewer #1 (Remarks to the Author):

This is an interesting study that integrates the development of CRISPR-mediated methods to tag low abundance and/or transiently expressed genes with some cell biology insight into mechanisms of circadian rhythms using the fluorescently-tagged clonal lines they developed. They generated knock-ins of fluorescent proteins (mClover3 or mScarlet-I) at two clock genes, *Per2* and *Cry1*, using a clever selection strategy that offers several advantages for targeting clock genes, which are typically expressed transiently at specific times of the day at low levels. Importantly, their methods allow removal of the positive selection cassette inserted downstream of the gene by transfecting with a Cre expression plasmid, restoring endogenous 3'-UTRs, which are frequently known to play a role in post-transcriptional regulation of gene expression. These methodological developments should be of broad interest to those studying non-abundant genes.

After thorough screening for both singly or doubly-labeled cells (e.g., cells with CRY1-mClover3 and PER2-mScarlet-I), they studied cellular circadian dynamics of the two clock proteins, demonstrating circadian changes in their abundance and nuclear localization that were largely expected based on prior data. The omission of key citations that previously established the delayed phase of *Cry1* expression and the importance of this in circadian rhythms detracts a bit from the conclusions of this study by not putting them in the context of what is already known in the field. However, the unique tools developed here allowed for the observation that CRY1 protein abundance is likely in ~6-8-fold excess relative to PER2 in intact cells, calling into question an over-simplified canonical model of circadian repression that all of the CRY and PER proteins are bound in large complexes with one another. The cellular tools and careful quantification of CRY1 and PER2 levels in cycling cells here contribute to an emerging model where the delayed expression of CRY1 gives rise to its unique role in the mammalian circadian clock (see below for details). With the inclusion of several citations to properly attribute earlier observations of delayed CRY1 expression, nuclear accumulation, and function in the clock, these data provide new evidence of, and new tools to explore, the delayed role of CRY1 in the clock. The questions raised here about the dependence of CRY1 on PER2 for nuclear entry and its activity outside of (and after) the large PER-CRY complexes that dominate the early part of repression should be of interest to the circadian community. Overall, there are a few concerns and missed opportunities to put these data in the context of prior data and models, but these should be easily addressable.

Major:

1. The data in Figure 2I (and elsewhere) provide strong evidence for a significant delay in the peak of CRY1 entry into the nucleus; however, there is no citation of the publication that first discovered the phase-delayed expression of *Cry1* (Ukai-Tadenuma et al. (2011) *Cell*) or several that followed, demonstrating that the delayed expression encoded by the *Cry1* promoter is essential for reconstitution of circadian clocks in cells and in vivo (Khan, S. et al. (2012) *J Biol Chem*, Li, Y. et al. (2016) *BBRC*, Rosensweig, C. et al. (2018) *Nat Comm*, and Maywood, E. et al. (2018) *PNAS*).

The Reviewer is right that we should have much better described previous research demonstrating both the delay in Cry1 mRNA and protein expression as well as the necessity of this delay for circadian oscillator function. While these studies used population sampling to come to these conclusions, our study, for the first time, shows this delay in single cells. In the new version of the manuscript we expanded the corresponding part of the Introduction (and added the relevant citations) as well as referenced to these findings in the Discussion.

2. The language surrounding 'the current model of the oscillator' seems a bit oversimplified. We agree that language surrounding the prevailing model tends to be a bit dogmatic, but there are experimental data from mice supporting a unique, PER-independent role for CRY1 late in repression going back as far as Stratmann, M. et al. (2010) *Genes & Dev* and Koike, N. et al. (2012) *Science*, and explicit models have been put forth since Ye, R. et al. (2014) *Genes & Dev*, Gustafson, C. and Partch, C. (2014) *J Mol Biol*, and Merbitz-Zahradnik, T. and Wolf, E. (2015) *FEBS Letts*. Presenting the current state of knowledge as though there is just one model, especially in the face of your data, seems like it incompletely addresses the state of the field.

Again, we absolutely agree with the Reviewer. In the new version of the manuscript, we now explicitly describe these findings in the Introduction (and Discussion) that emphasize the special role of CRY1 as a late repressor.

3. The text in the Discussion that "In mouse liver, biochemistry data suggest that CRY and PER proteins almost exclusively coexist in huge cytosolic and nuclear complexes suggesting common regulation" also does not capture the full breadth of literature on this topic. First, we encourage the authors to look at western blots in Supp. Fig. S1 for the paper they cite here (Aryal, R. et al. (2017) *Mol Cell*), which indeed show a considerable amount of CRY1 that is not sequestered in large PER-CRY complexes, particularly at later phases in the cycle. Second, this statement also neglects to cite biochemical ChIP-Seq data from mouse livers from the Takahashi lab (Koike, N. et al. (2012) *Science*) that also shows the apparent PER-independent recruitment of CRY1 to CLOCK:BMAL1 late in the repressive cycle. Presenting a more holistic view of the field would strengthen the clock-specific contributions of this manuscript.

Again, we should have cited the Koike et al. paper that nicely shows the PER-independent role of CRY1 as a late repressor. In addition, the Reviewer is correct that monomeric CRY1 is detected in considerable amounts (maybe more than a "minor" amount - as we stated in the first version of our manuscript) also suggesting a complex-independent CRY1 role. Nonetheless, the 5-10 excess of nuclear CRY1 over PER2 (which we now confirmed with independent technique – see answer to Reviewer#3) is still not easily compatible with the data shown in Aryal, R. et al., where between CT8 and CT20 most of CRY1 seems to be in a large complex. Between CT0 and CT4 (the phase, when CRY1 co-occupies CLOCK/BMAL1 binding sites in the genome) Aryal's data point to about 50% of CRY1 protein being present as monomers, while the rest can still be detected within the large PER-dependent repressor complex. In the new version of our manuscript, we specifically point to these data that show substantial amounts of CRY1 monomers in the late repression phase.

Minor:

1. In supplementary figure 1G, the bottom panel was not labeled like the upper panel (i.e., PCR: PER2) and it is not clear where the arrows are pointing to indicate positive clones—is the PER2-FP arrow pointing to the ladder of bands in the middle lane as the positive clone? Perhaps something is off in the alignment of the arrowheads?

We thank the Reviewer for this observation. We added the missing label 'PCR: PER2'. Admittedly, the respective bands were not easy to recognize. Thus, we added additional arrowheads within the panel to mark the PCR products resulting from knock-in alleles, whose sequences we all verified by Sanger sequencing.

We also would like to point out that many of the bands in heterozygous clones most likely are PCR-artifacts and can be explained by the formation of heteroduplexes between wild-type and knock-in PCR products, as we described in Material and Methods (chapter: 'nucleic acid isolation and PCR').

2. It is stated in Supplementary figure 4A-D that the data represent 4 independent runs from 2 experiments, but $n = 2$ points are shown for period and amplitude and lack statistical analyses. Were statistical tests run on the full dataset to support the claim that there is no difference between the clonal lines and wild-type cells?

For the data shown in **Supplementary Figure 4**, we performed two completely independent experiments ($n=2$; i.e. two experimental days). In each experiment we independently recorded two cell culture dishes for each clone (described in the legend: mean + SD of four individual, detrended time-series resulting from two independent experiments). We consider the two time-series and extracted data (period, amplitude) from one experiment as technical replicates and therefore did not test statistical difference for $n=2$ independent experiments. If, however, we regard each time-series as an independent sample (resulting in $n= 4$ for each clone) and compare amplitude and period of the clones using one-way ANOVA, we do not find significant differences.

In addition, we and others recently showed that periods from individual U-2 OS (and other cell line) clones can considerably and robustly vary and deviate from the average period of the parental culture (Li et al., 2020; Nikhil et al., 2020). Thus, a slightly longer or shorter period would not necessarily point to an integration-dependent alteration of the molecular clockwork but could as well be caused by clonal differences (or a combination of both).

Our intention was to show that the circadian rhythm in our clones is intact and periods and amplitudes are not deviating largely from the expected circadian range, as it is seen e.g. in CRY1 or PER2 knock-out cells (Börding et al., 2019; Liu et al., 2007). However, from our data we cannot completely rule out that the fluorophore knock-in into PER2 and/or CRY1 may have subtle effects on the circadian rhythm parameters.

3. Although there is valuable information in tracking CRY1 and PER2 fluorescence

immediately after synchronization by dexamethasone (e.g., Fig. 2A-D but particularly PER2 expression in 2D), it is common practice to omit or gray out data from the first 12-24 hours to focus on expression driven later by circadian oscillations. Could you annotate this first peak or remark on this somewhere for those who are not in the circadian field?

We thank the Reviewer for the advice. We overlaid the graphs in **Fig. 2b-e** and **Fig. 3c,d** with gray boxes for the first 24 hours and marked them as potential acute dexamethasone response.

4. The PER2-mScarlet-I cells show robust circadian expression of the tagged protein, which may be due to its overall higher fluorescence compared to the PER2-mClover3 clone (~10X lower)...

The Reviewer is correct. The fluorescence values after background subtraction of the PER2-mScarlet-I clone are much higher than those of the PER2-mClover3 clone. This is in part due to a higher expression of PER2-fusion proteins in these cells. Indeed, our copy-number analysis revealed that the PER2-mClover3 clone possesses one additional wild-type allele (see new **Supplementary Fig. 1i**). However, the fluorescence units are arbitrary units and cannot directly be compared between the different fluorescence channels, since they depend on instrumental settings (laser intensity, detector sensitivity, etc.)

... However, this is not readily visible in either the PER2-mClover3 or CRY1-mScarlet-I HCT-116 cells shown in Supp. Fig. 5. Is the punctate staining here also due to auto-fluorescence?...

Yes, as shown for the U-2 OS cells in **Supplementary Fig. 1c-e**. To clarify this further (at least for U-2 OS cells), we also subjected our cells to fluorescence lifetime imaging (FLIM) using a two-photon microscope. Interestingly, the perinuclear auto-fluorescence has a much shorter lifetime than mClover3 and can thus be distinguished by FLIM (new **Supplementary Fig. 1f**). See also our detailed answer to Reviewer#3.

...These data do not seem to provide strong evidence that recapitulates the U-2 OS data.

We thank the Reviewer for this comment. To support our claim that HCT-116 double knock-in cell data recapitulate our findings in U-2 OS cells, we should have analyzed the data as vigorously as for U-2 OS cells. Thus, we analyzed time-series of ten individual HCT-116 double knock-in cells (including the four previously shown). Nuclear expression of both proteins were rhythmic in nine out the ten cells, with average periods of 23.6 hours - similar to U-2 OS cells (new **Supplementary Fig. 5c,d and f**). As in U-2 OS cells, the amplitude of PER2 nuclear expression rhythms is larger than that of CRY1 (new **Supplementary Fig. 5e**), although the latter seems to be larger in HCT-116 cells compared to U-2 OS cells (new **Supplementary Fig. 5g**). In addition, we also observed the phase difference between nuclear PER2 and CRY1 rhythms in HCT-116 cells (new **Supplementary Fig. 7c**). Together, this

indicates that clock protein dynamics is very similar in HCT-116 cells compared to U-2 OS cells.

5. While it is expected that the intensity of the first peak of fluorescence in doubly tagged cells (CRY1-mClover3 and PER2-mScarlet-I) would be the highest in Fig. 3C, it is curious (and interesting) that the second is of quite low fluorescence intensity, while the 3rd is much higher. This is not commonly observed with clock protein-luciferase fusions (i.e., PER2::LUC and Fig. 4 here)—has this been observed with any of the prior studies fluorescently tagging clock proteins? Is there any explanation for this?

We thank the Reviewer for the question. Fig. 3b,c show data of one individual cell clone. Data from single cells are inherently rather noisy, and the amplitude/period/peak expression values can vary quite a lot from cycle to cycle (see also (Bieler et al., 2014) and (Yang et al., 2020) Fig. 3E). However, when looking at the collective data from several cells (Fig. 2e, 3d), we did not observe that the second peak after dexamethasone treatment is consistently lower than the third.

To quantify this, we reanalyzed the peak expression of all six presented time-series collections (as shown in Fig. 2e, 3d) and did not find a significant difference between 2nd and 3rd peaks' fluorescence intensities (see Figure below).

Fig. R1. Comparing peak expression of PER2 (left panel) and CRY1 (right panel) on day 2 and day 3 after synchronization by dexamethasone. p-values by Wilcoxon matched-pairs signed rank test, n= 91 (PER2) and 92 (CRY1), resp.

6. The suggestion in the Discussion that “it is also conceivable that a single PER2 protein may be able to shuttle multiple CRY proteins, either at a time or in a sequential manner” does not seem to be plausible given that each PER protein has a single well-studied and high-affinity CRY-binding site. Are you suggesting that there is another cryptic CRY-binding site on PER2, or that dynamic nuclear shuttling of PER2-Venus is somehow correlated with the release of CRY proteins in the nucleus for the sequential transport of multiple CRY proteins? While this could be true, the data here fall far short of probing anything like this and it seems like a bit of a reach, even for a discussion.

The Reviewer is correct that possible mechanisms that may explain the abundance differences between nuclear CRY1 and PER2 are currently mere speculation and not (yet) backed by experimental data. While we do not suggest a cryptic CRY1-binding site on PER2 (thus, we deleted the phrase 'either at a time'), we think it is not completely implausible that the binding affinity between PER2 and CRY1 (that has been determined with non-phosphorylated purified proteins) depends on localization (and circadian phase). Since previous data argue that (i) CRY1 nuclear localization depends on the presence of PER proteins and (ii) monomeric CRY1 has a PER-independent role as a late repressor, we speculate that nuclear CRY1 can dissociate from complexes with PER proteins (maybe in a phosphorylation state-dependent manner). Whether this would free PER proteins to shuttle additional CRY proteins is indeed currently unclear.

To conceptualize possible mechanisms, we performed mathematical modeling experiments, in which we explicitly incorporated the nuclear dissociation of PER2 and CRY1. We explored the parameter space of the PER2:CRY1 loop and found a set of parameters that reproduced our experimental findings, namely (i) a circadian period, (ii) a higher abundance of nuclear CRY1 compared to PER2, (iii) a larger amplitude of PER2 rhythms, and (iv) a delayed peak of nuclear CRY1 compared to nuclear PER2 rhythms. This model now provides an alternative explanation for the high abundance difference between PER2 and CRY1 in the nucleus. Firstly, it predicts that the half-life of nuclear CRY1 is higher than that of nuclear PER2, which might explain the abundance difference. Secondly, it predicts that the association of cytoplasmatic PER2 and CRY1 occurs at a higher rate than the dissociation and, conversely, in the nucleus, the dissociation of the PER2:CRY1 complex should occur at a higher rate than the association. This would drive the net PER2:CRY1 flux towards the nuclear monomers.

We would like to discuss these various possibilities in our manuscript and suggest experimental approaches, how to test these predictions, even though we agree with the Reviewer that experimental evidence is only sparse (but see data on CRY1 and PER2 half-life (Hirano et al., 2014, 2017; Jang et al., 2018; Rosensweig et al., 2018; Siepka et al., 2007; Smyllie et al., 2016)).

We present our new modeling results in **Fig. 5** and **Supplementary Fig. 9**.

Reviewer #2 (Remarks to the Author):

In accordance with the Editor's request, this Reviewer provides a review with aspects related to CRISPR gene editing. I leave the evaluation of other aspects of the manuscript to other referees.

In the context of genome editing, the authors created knock-in cells carrying fluorescence reporter genes at the CRY1 and PER2 loci. The authors' method includes a floxed positive selection cassette and a negative selection cassette at the outside of the homology arm. This approach should be particularly useful for tagging the endogenous genes with low expression.

The authors describe their method as a "new strategy"; however, there is no technologically novel point. The insertion of fluorescence reporter gene nearby the stop codon is a common maneuver for the visualization of endogenous gene expression. Floxed positive selection cassette and the negative selection to avoid the random integration are also the well-characterized strategies. Coexpression of i53bp was previously reported, too. The knock-in strategy itself is also a standard HDR-mediated gene insertion. This reviewer thinks this report does not contain sufficient technological advance to consider publishing in Nature Communications unless there are sufficient novelty and impact in other aspects such as the dynamics analysis of clock proteins.

We agree with the Reviewer that none of the individual elements of our knock-in strategy is particularly new. For example, the combination of positive and negative selection is standard in generation of transgenic mice. However, to our knowledge, the *combined* usage of both is rarely reported when somatic knock-in cells are generated.

From personal discussions within the circadian field, we learned that many researchers failed to produce CRISPR/Cas9-mediated knock-in cells for circadian clock proteins. Most colleagues did not implement enrichment strategies, probably motivated by reports of highly efficient one-step knock-in generations, which in our experience is rather an exception (e.g. in HEK293 cells or stem cells). In addition, several enrichment strategies do not work faithfully when targeting lowly or transiently expressed genes (as it is the case for many circadian clock genes).

Thus, by combining several well-characterized elements, we present a thorough and robust selection strategy to enrich for correctly edited cells, which may help other researchers to overcome difficulties during CRISPR mediated targeted knock-in generation.

Again, the Reviewer is correct that neither of the elements is new. We also do not wish to claim that the described combination has never been applied before. Therefore, we substituted the word 'new' by 'efficient' or 'robust'.

Additional specific comments:

1. Off-target analysis should be fully performed. Treatment of i53bp and co-introduction of three sgRNAs may increase the potential risk.

We agree with the Reviewer that thorough analysis of potential Cas9 off-target mutations is an important point. Therefore, we performed additional experiments and analyses.

While analysis of single potential off-target sites (e.g. by PCR and Sanger sequencing) is of very limited information on a clonal basis, next generation sequencing offers the possibility of analyzing thousands of those sites in parallel. Since indels in exonic regions are more likely to be detrimental for protein function (e.g. by inducing frame-shift or nonsense mutations) than indels in intronic and intergenic regions, we performed whole exomes sequencing (WES) of multiple clones to cover the majority of the most relevant genomic regions with reasonable efforts. Specifically, we analyzed WES data for the four single knock in clones shown in **Fig. 2**; for two CRY1-mClover-3/PER2-mScarlet-I double knock in clones, namely #4

and #6 (**Supplementary Fig. 6**), the latter of which was used to produce the data shown in **Fig. 3** and **Fig. 4**; and for the founding wild-type cell population.

We detected between ~20,400 and 22,600 indels per clone, 2-3 % of which mapped to protein coding regions, 1.3–1.4 % to ncRNA coding regions and the remaining to intronic (69-72%), UTR (4.8-5.2%) and intergenic regions. From these, we filtered out all indels that were also detected in the founding wild-type population or one of seven previously sequenced wild-type clones (Nikhil et al., 2020), as these indels are unlikely to be a product of Cas9 activity. This resulted in ~4,000-6,000 uniquely detected indels per clone.

Finally, we analyzed, which of these indels are located in direct vicinity (within 20 bp) to predicted off-target sites of the four sgRNAs (1 targeting *CRY1*, 3 targeting *PER2*) used to generate the knock-in clones. While almost 1,600 predicted off-target sites from four different prediction tools map to sequenced genomic regions, none was associated with any of the identified unique indels. Thus, the observed indels are unlikely to be the result of off-target effects, but rather are due to clonal variations or differential coverage during exome sequencing.

In summary, we did not find any evidence for Cas9-mediated indel formation in exonic regions. However, since WES mainly covers exonic regions, we cannot rule out the occurrence of Cas9 off-target effects in other regions of the genome.

We included these findings to the main text of the manuscript. In addition, we provide a detailed description of the analysis pipeline and the results (new **Supplementary Note 1**). We are confident that these results strengthen the significance of our cell biology data and thank the Reviewer for the suggestion.

2. Genotyping experiments are not sufficient. The authors only performed standard PCR and sequencing. Southern blotting and/or copy number analysis using ddPCR quantification should be additionally performed to demonstrate the non-existence of random integration and the proper on-target genotype.

We thank the Reviewer for these suggestions. To address this point, we employed two strategies:

1. Copy number variation analysis by ddPCR

As suggested by the Reviewer, we determined copy number of the transgenic fluorophores (mScarlet and mClover3) by ddPCR, relative to regions of the targeted gene (*CRY1* or *PER2*) outside of the homology region.

For the *CRY1* locus of the single and double KI cells, ddPCR confirmed the expected ratio of fluorophore sequence to targeted gene, *i.e.* 1:1 for the presumed biallelic knock-in (*CRY1*-mClover3) and 1:2 for mono-allelic knock-ins (new **Supplementary Fig. 1i**). Similar results were obtained for the *PER2* locus in the double knock-in cells (new **Supplementary Fig. 6d**). For *PER2* in single knock-in clones, the copy number analyses yielded some unexpected results:

- The ratio of mClover3 to *PER2* was 1:3, pointing towards a third copy of the *PER2* gene in this one clone (new **Supplementary Fig. 1i**). The copy ratio of *PER2:CRY1* in this clone was 3:2, supporting the hypothesis of a third *PER2* allele. To follow up on this, we evaluated the WES data for this region and found roughly twice as many reads of the wild-type-with-indel sequence (as seen by Sanger sequencing, **Supplementary Fig. 2c**) as of the knock-in sequence. Thus, our interpretation is that in this one clone, the *PER2* locus or chromosome 2 as a whole are present with three copies. A duplication either occurred beforehand, followed by identical indel formation on two alleles, or between knock-in and clonal selection by duplicating the wild-type-like allele. Indeed, the genome of U-2 OS cells is described to be rather instable ((Ozaki et al., 2003), ATCC U-2 OS cell line characteristics (<https://www.atcc.org/products/all/HTB-96.aspx>)). This may also explain in part, why the signal to noise ratio of this clone was inferior to the *PER2*-mScarlet knock-in clone as seen in **Fig. 2**. We updated the main text with this new information.
- In the first version of our manuscript, we (as it turns out, prematurely) interpreted the absence of a wild-type band in *PER2*-mScarlet-I knock-in cells during out-out PCR as indication for a biallelic knock-in. However, copy number analysis by ddPCR revealed an mScarlet-I:*PER2* ratio of 1:2, indicative of a mono-allelic knock-in (new **Supplementary Fig. 1i**). We speculated that this discrepancy might be due to a larger Cas9-mediated deletion on the second allele, including one of the PCR primer binding sites. Indeed, by using primer with more upstream and downstream annealing sites, we obtained a second PCR product, which was much shorter than calculated. Sequencing revealed a deletion of 2138 bp upstream to 784 bp downstream of the stop codon. Thus, this *PER2*-mScarlet-I single knock in clone represents a mono-allelic knock-in, with the second allele bearing a deletion of the last exon's coding region rather than a biallelic knock-in as proposed previously.

We updated the text and the respective figures (**Fig. 4d** and **Fig. 2d**).

Importantly, the obtained ratios also demonstrate that no additional copy of the fluorophores are present in the genome of any tested clone, which would have resulted in a higher than expected fluorophore-to-target gene ratio. This is in line with our findings that fluorescence decreased to near-background levels upon knockdown of the corresponding fusion protein partner mRNAs (*i.e.* *CRY1* or *PER2*, respectively, **Fig. 1f** and **Supplementary Fig. 3**).

2. Analysis of WES data for hints of random integration events.

We took advantage of the obtained paired-end WES data and analyzed it for the presence of any donor plasmid sequence in exonic regions. To this end, we remapped the WES reads to a modified reference genome that additionally contained the donor plasmid sequences. In case of integration, we expected to find discordant sequencing reads that can be mapped with one end to the sequence of the donor plasmid and with the other end to that of the exon sequence, or paired reads, one of which can be mapped to the vector sequence and the other to an exon. Overall, we found only little if any evidence for random integration of the vector into exonic regions of the genome. There are three peculiarities, though, which - although very unlikely to affect our biological results - we wish to share with the Reviewer:

- In one clone (*CRY1*-mClover3), we observed several reads spanning a part of the CD4 cDNA sequence of the negative selection cassette fused to a part of the 3'-homology region of *CRY1* (intron 12 of *CRY1*). The CD4 part of the sequences could be mapped exclusively to the vector as they covered several exon-junctions without intronic sequences. A possible explanation is that this resulted from a recombination of the donor vector, which was then randomly integrated into a *non-exonic* region (integration in an exonic region would have likely resulted in further discordant read(-pairs) at the integration site). The sequence is covered by exome sequencing since the CD4-part can hybridize with the selection probes.
- In addition, one *single* read in the *PER2*-mScarlet-1/*CRY1*-mClover3 double knock-in clone #4 aligned to the CD4 coding region as well as to the MYOF gene on chromosome 10. Since the respective MYOF region was covered by >30 further reads all showing a regular pattern, we do not regard this as (strong) evidence for random vector integration at this site, as vector integration would be expected to manifest in a more substantial proportion of reads (*e.g.* roughly 50 % for a two-allelic gene).
- Furthermore, we detected *four* read pairs of which one read mapped to the donor vector and the other read to an unrelated genomic region. Of note, all of the concerned vector regions are also present in the genome. I.e., the suspicious read pairs can also be seen as read pairs that map to different genomic regions. With ~0.3 percent of all read-pairs of the dataset (*i.e.* >100.000 per clone) pairing to different chromosomes, this is a rather common phenomenon, which can either be an artifact of the library preparation or result from genetic rearrangement. Yet, we cannot rule out the possibility of vector integration at these genomic sites. To our knowledge, the related genes *TEX35*, *CFAP44*, *HSD17BS* have not been linked to the regulation of circadian rhythms; hence, it is rather unlikely that alterations in one allele of any of these genes would strongly affect the results of this study.

To summarize, we did not find strong evidence for random integration of any vector part into exonic regions of the genome. Again, we cannot exclude random integration of donor plasmid sequences into intronic or intergenic regions.

We added the key findings of this analysis to the main text. Additionally, we provide a thorough description of the strategy and results in the new **Supplementary Note 1**. In addition, the sequencing data can be assessed at <https://www.ncbi.nlm.nih.gov/sra/PRJNA677859>.

3. Regarding the last sentence of the Abstract, triple knock-in was not supported in this study at all.

The Reviewer is correct. We changed the respective sentence in the abstract.

Reviewer #3 (Remarks to the Author):

This study by Gabriel et al on “Live-cell imaging of circadian clock protein dynamics in CRISPR-generated knock-in cells” represents one of the earliest attempts to generate combinations of fluorescent clock protein reporters at their endogenous loci in the same cells using CRISPR technique. This approach is intrinsically challenging because 1) the

potential “off-target” nature of CAS9 by random integration of the donor sequence into the genome; 2) the low abundance (hence low signal) of clock transcription factors. The authors have taken great caution in generating and characterising CRISPR knock-in cell clones, with clever tricks (including both positive and negative selection process) to minimise the two problems mentioned above. They have targeted two highly rhythmic clock factors CRY1 and PER2 and tagged them with various FP and LUC tags. To make the results directly comparable, they also generated double knock-in cells with PER and CRY tagged by different FPs.

These efforts are paid off in that several new insights have been obtained including the visualisation of both PER2 and CRY1 in the same cells, with the most exciting discovery being the nuclear localisation of CRY1 throughout the circadian cycle without clear circadian gating (contrary to current models based on low resolution biochemical analysis). In addition, CRY1 is phase delayed by ~5 hours as compared to PER2 in the same cells (based on both FP and LUC reporters). Although PER2 has been extensively studied in fruit flies and mice (including PER2::VENUS and PER2::LUC), there have been no critical studies of tagging endogenously expressed CRY1 proteins using real-time reporters in cells, and nor studies to tag both PER and CRY in the same cells. As such, this paper represents both technological and biological advances in our understanding of circadian cell biology.

The experiments are overall very well designed with proper controls and validations in different cell lines (U2OS, HCT), different synchronisation methods (dex vs temperature) and different types of reporters (FP vs LUC). Functional tagged clock proteins were confirmed by sequencing of junctions, shRNAs and by sustained circadian rhythms in cells. There are a few weaknesses that could be improved by in depth discussion, analysis or additional experiments:

1) It seems a missed opportunity that the authors did not attempt to build a mathematical model based on the semi-quantitative imaging data (phase/amplitude/abundance difference between PER and CRY), despite being experts in the clock modelling field.

Encouraged by the Reviewer’s comment, we performed mathematical modeling to conceptualize our findings. In the model, we explicitly incorporated the nuclear dissociation of PER2 and CRY1. We explored the parameter space of the PER2:CRY1 loop and found a set of parameters that reproduced our experimental findings, namely (i) a circadian period, (ii) a higher abundance of nuclear CRY1 compared to PER2, (iii) a larger amplitude of PER2 rhythms, and (iv) a delayed peak of nuclear CRY1 compared to nuclear PER2 rhythms. This model now provides an alternative explanation for the high abundance difference between PER2 and CRY1 in the nucleus. Firstly, it predicts that the half-life of nuclear CRY1 is higher than that of nuclear PER2, which might explain the abundance difference. Secondly, it predicts that the association of cytoplasmatic PER2 and CRY1 occurs at a higher rate than the dissociation and, conversely, in the nucleus, the dissociation of the PER2:CRY1 complex should occur at a higher rate than the association. This would drive the net PER2:CRY1 flux towards the nuclear monomers.

We present our new modeling results in **Fig. 5** and **Supplementary Fig. 9**.

2) All the PER2-FP reporters showed rather weak signals, limiting their utility. This is partly due to the low abundance of PERs. Is it also possible that the large PER proteins somehow interfere with the folding or maturation time of FP tags?

Yes, in principle it is conceivable that fusion protein partners influence each other's properties. To lower the risk of adverse crosstalk, the fusion protein parts in our system are separated by a linker sequence of 21 amino acids. Although, we cannot rule out the possibility of a crosstalk, we think it is unlikely, because of the following: If PER2 fusion increases a fluorophore's maturation time, apparent peak expression of the PER2 fusion protein would be delayed and maturation time dependent. However, irrespective of whether CRY1 is fused to mScarlet-I and PER2 to mClover3, or vice versa, or whether both are fused to luciferase, the observed phase relation between peak expression of CRY1 and PER2-fusion proteins is constant as also observed by others (e.g. Narumi et al., 2016; Wang et al., 2017). From this, we conclude that it is rather unlikely that low PER2-FP signals are due to PER2 altering the maturation time of the fused fluorophores.

At the trough level, it is very hard to tell whether there is any real signal above background. The authors stated: "however, a reliable discrimination from auto-fluorescence and background signals was not possible". Have the authors considered using spectral lambda imaging and linear unmixing to eliminate auto-fluorescence?

We thank the Reviewer for this question. The cited statement related to our difficulties to detect/quantify fluorophore levels in the cytoplasm, where strong auto-fluorescence is a major hurdle for imaging low levels of fluorescence. In contrast, auto-fluorescence was not as strong in the nucleus and thus no major issue for imaging and quantification. Here, *PER2*-mScarlet-I trough levels were not distinguishable from the background (as shown in **Supplementary Fig. 4g**). This is not surprising, as PER2 protein levels have been reported to be extremely low or even undetectable at the trough (Narumi et al., 2016; Wang et al., 2017). To our knowledge, spectral imaging cannot improve the signal-to noise ratio of very low signals in the absence of auto-fluorescence.

Motivated by the Reviewer's comment, we performed simultaneous spectral imaging on a multi-detector scanning confocal microscope to better quantify cytoplasmic fluorescence, *i.e.* simultaneous excitation/emission detection with 405 nm/425-475 nm (blue light channel), 488 nm/500-550 nm (green light channel) and 561 nm/570-620 nm (red-light channel)) with the aim to eliminate auto-fluorescence signals by post-processing.

We found that perinuclear punctuate auto-fluorescence is mainly present in the green and red channel (**Fig. R2**). Spectral unmixing can eliminate a substantial amount from the green channel (**Fig. R2A**), but only to a small extent from the red channel (**Fig. R2B**).

In addition, filamentous autofluorescence is present in the blue and green channel and can be observed throughout the cytoplasm. Spectral unmixing removed a fraction of this, but the results did still not allow a reliable distinction of low signals from auto-fluorescence in the cytoplasm.

Figure R2: Spectral imaging of *CRY1*-mClover3 cells (**A**) and *PER2*-mScarlet-I cells (**B**). Overlay and spectral unmixing was done in ImageJ.

Overall, we see some limits of this approach for our experiments.

(1) Spectral imaging is prolonging the time or intensity of laser or LED light exposure, thus increasing both the risk of photobleaching and cytotoxic effects. Since we usually record over the course of days, we always try to restrict illumination time to the necessary minimum.

(2) From a practical view, we can only do simultaneous spectral imaging on a scanning confocal microscope in our facility. Confocal imaging is also associated with prolonged sample illumination and we experienced strong bleaching of our already low signals. Therefore, we used a wide-field microscope for all our time-series experiments.

(3) Since much of our signals are close to background, we are afraid that spectral unmixing may create strong artefacts in our data.

In another attempt to circumvent the above limitations, we subjected our cells to fluorescence lifetime imaging (FLIM) using a two-photon microscope. Interestingly, the perinuclear auto-fluorescence has a much shorter lifetime than mClover3 and can thus be distinguished by FLIM (new **Supplementary Fig. 1f**). However, the sensitivity of the two-photon microscope was lower than that of our wide-field microscope. As a result, only *CRY1*-mClover, but not *PER2*-mScarlet was detectable, and only after averaging over prolonged imaging time. As before, we did not observe cytoplasmic *CRY1*-mClover signals above background.

We added these new data as a new **Supplementary Fig. 1f**, which further suggest that the punctuate fluorescence signals are cellular auto-fluorescence

Supplementary Fig. 1f: Fluorescence lifetime imaging (FLIM) of *CRY1*-mClover3 knock-in cells reveal different fluorescence lifetimes (τ) of specific nuclear and auto-fluorescent perinuclear signals. Scale bar: 30 μm .

3) One of the main conclusions is the 6-10 fold difference in abundance between *PER2* and *CRY1*, based on fluorescence intensity. Given that *PER* and *CRY* are very different proteins (molecular weights, PTMs, structures), the FP signal intensity is likely to be skewed by such intrinsic differences (even when the same FP was tagged on different proteins). Therefore, this conclusion needs to be validated by additional quantitative imaging (for example, measuring # of molecules in a confocal volume) or biochemical methods (Western Blot taking advantage of the His flag tag in the constructs).

We thank the Reviewer for this excellent suggestion. To determine the ratio of *PER2* and *CRY1* by a complementary method, we performed fluorescence correlation spectroscopy (FCS) to estimate the concentration of *CRY1*-mClover3 and *PER2*-mScarlet-I fusion proteins in individual double knock-in cells at the time of *PER2* peak expression. Importantly, FCS estimates molecule numbers in a small confocal volume based on diffusion events and as such is independent of a fluorophore's brightness.

Using FCS, we determined the molecular concentrations of *PER2*-mScarlet-I and *CRY1*-mClover3 in nuclei of double-knock in cells 34-36 h after dexamethasone synchronization (2nd peak of *PER2* expression). We analyzed fusion-protein concentration in five single

confocal volumes per nucleus and calculated the mean to estimate the cell's expression level. We appended the corresponding results to **Fig. 4** (new **Fig. 4f-h**).

Over all measured confocal volumes, we obtained a mean nuclear concentration of 4.5 nM for CRY1-mClover3 and of 0.5 nM for PER2-mScarlet-I (new **Fig. 4f**, medians 2.5 nM and 0.2 nM, respectively), and similar results were obtained after calculating the mean values *per cell* (new **Fig. 4g**). Note, these values refer to the contribution of only one (knock-in) allele.

Thus, we obtained median ratios of CRY1 to PER2 of 8-10 fold (new **Fig. 4h**), which is in good agreement with the 8.5 fold ratio from fluorescence image quantification. Interestingly, we obtained higher diffusion constants for CRY1-mClover3 ($2.3 \pm 0.9 \mu\text{m}^2/\text{s}$) than for PER2-mScarlet-I ($1.0 \pm 0.8 \mu\text{m}^2/\text{s}$), which suggests that a substantial part of CRY1 is not integrated into large protein complexes (new **Fig. 4i**).

It is worth mentioning that, although FCS-based measurement of concentrations is rather insensitive to absolute fluorophore brightness, it has its own limitations, which we discuss in the Materials and Methods section. For example, FCS can only measure the mobile fraction of molecules, thereby underestimating the real concentration especially for proteins with a large immobile fraction, which is not a limitation for the conventional (semi-)quantitative fluorescence measurements that we did before.

In summary, a complementary method recapitulated the observed PER2 to CRY1 abundance ratios supporting our initial finding that CRY1 is much more abundant in the nucleus of U-2 OS cells than PER2.

4) A key part of the result section has been dedicated to the generation and validation of the CRISPR cell lines. Even though these methodologies are important for peers in future efforts, it somehow dilutes the key biological insights. I wondered whether some of these could be moved to Methods section?

We agree with the Reviewer that our manuscript constitutes a combination of technology description and biological insights and we concede that each reader has his/her own particular interest. Based on our personal discussions with many colleagues in the field, who were interested in both CRISPR/Cas9 technology application as well as in circadian clock mechanisms, we would prefer to keep the balance between these themes as it is. Importantly, we moved all new methodological data (WES, modeling details) into the Supplement to not further dilute the biological insights. In addition, we tried to shorten the (original) methodological part as far as we could (although, admittedly, due the new validation data, it did not become much shorter).

5) U2OS is an osteosarcoma-derived cell line. It would be interesting to discuss whether the conclusions drawn in U2OS would be broadly applicable to primary normal cells and the SCN.

We thank the Reviewer for this advice. In **Supplementary Note 3**, we discuss the value and limits of the U-2 OS cell line as a model system to study the molecular oscillator, and refer to it in during the main discussion. Note, that all major conclusions from U-2 OS cells could be recapitulated in HCT-166 cells (which admittedly are also cancer-derived cells; **Supplementary Fig. 5**).

6) In discussion, “This delay is also present on whole-cell protein level (Fig. 4C and 28) indicating that in single cells circadian nuclear accumulation of CRY1 and PER2 mainly reflects the circadian expression of those proteins rather than being the consequence of circadian gating in nuclear appearance.”. I think degradation of these repressors should play a role too, not just circadian expression.

With circadian "expression", we actually meant "protein abundance" as a result of production and degradation. We admit that the term expression is a bit ambiguous. To make this more clear to the reader, we changed 'expression' to 'expression level'. We thank the Reviewer for this comment.

References

- Bieler, J., Cannavo, R., Gustafson, K., Gobet, C., Gatfield, D., and Naef, F. (2014). Robust synchronization of coupled circadian and cell cycle oscillators in single mammalian cells. *Mol. Syst. Biol.* *10*, 739–739.
- Börding, T., Abdo, A.N., Maier, B., Gabriel, C., and Kramer, A. (2019). Generation of human CrY1 and CrY2 knockout cells using duplex CRISPR/Cas9 technology. *Front. Physiol.* *10*, 1–9.
- Hirano, A., Kurabayashi, N., Nakagawa, T., Shioi, G., Todo, T., Hirota, T., and Fukada, Y. (2014). In vivo role of phosphorylation of cryptochrome 2 in the mouse circadian clock. *Mol. Cell. Biol.* *34*, 4464–4473.
- Hirano, A., Braas, D., Fu, Y.-H., and Ptáček, L.J. (2017). FAD Regulates CRYPTOCHROME Protein Stability and Circadian Clock in Mice. *Cell Rep.* *19*, 255–266.
- Jang, J., Chung, S., Choi, Y., Lim, H.Y., Son, Y., Chun, S.K., Son, G.H., Kim, K., Suh, Y.-G., and Jung, J.-W. (2018). The cryptochrome inhibitor KS15 enhances E-box-mediated transcription by disrupting the feedback action of a circadian transcription-repressor complex. *Life Sci.* *200*, 49–55.
- Li, Y., Shan, Y., Kilaru, G.K., Berto, S., Wang, G.-Z., Cox, K.H., Yoo, S.-H., Yang, S., Konopka, G., and Takahashi, J.S. (2020). Epigenetic inheritance of circadian period in clonal cells. *Elife* *9*.
- Liu, A.C., Welsh, D.K., Ko, C.H., Tran, H.G., Zhang, E.E., Priest, A.A., Buhr, E.D., Singer, O., Meeker, K., Verma, I.M., et al. (2007). Intercellular Coupling Confers Robustness against Mutations in the SCN Circadian Clock Network. *Cell* *129*, 605–616.
- Narumi, R., Shimizu, Y., Ukai-Tadenuma, M., Ode, K.L., Kanda, G.N., Shinohara, Y., Sato, A., Matsumoto, K., and Ueda, H.R. (2016). Mass spectrometry-based absolute quantification reveals rhythmic variation of mouse circadian clock proteins. *Proc. Natl. Acad. Sci. U. S. A.* *113*, E3461-7.
- Nikhil, K.L., Korge, S., and Kramer, A. (2020). Heritable gene expression variability and stochasticity govern clonal heterogeneity in circadian period. *PLOS Biol.* *18*, e3000792.

Ozaki, T., Neumann, T., Wai, D., Schäfer, K.-L., van Valen, F., Lindner, N., Scheel, C., Böcker, W., Winkelmann, W., Dockhorn-Dworniczak, B., et al. (2003). Chromosomal alterations in osteosarcoma cell lines revealed by comparative genomic hybridization and multicolor karyotyping. *Cancer Genet. Cytogenet.* *140*, 145–152.

Rosensweig, C., Reynolds, K.A., Gao, P., Laothamatas, I., Shan, Y., Ranganathan, R., Takahashi, J.S., and Green, C.B. (2018). An evolutionary hotspot defines functional differences between CRYPTOCHROMES. *Nat. Commun.* *9*, 1138.

Siepkha, S.M., Yoo, S.H., Park, J., Song, W., Kumar, V., Hu, Y., Lee, C., and Takahashi, J.S. (2007). Circadian Mutant Overtime Reveals F-box Protein FBXL3 Regulation of Cryptochrome and Period Gene Expression. *Cell* *129*, 1011–1023.

Smyllie, N.J., Pilorz, V., Boyd, J., Meng, Q.-J., Saer, B., Chesham, J.E., Maywood, E.S., Krogager, T.P., Spiller, D.G., Boot-Handford, R., et al. (2016). Visualizing and Quantifying Intracellular Behavior and Abundance of the Core Circadian Clock Protein PERIOD2. *Curr. Biol.* *26*, 1880–1886.

Wang, J., Mauvoisin, D., Martin, E., Atger, F., Galindo, A.N., Dayon, L., Sizzano, F., Palini, A., Kussmann, M., Waridel, P., et al. (2017). Nuclear Proteomics Uncovers Diurnal Regulatory Landscapes in Mouse Liver. *Cell Metab.* *25*, 102–117.

Yang, N., Smyllie, N.J., Morris, H., Gonçalves, C.F., Dudek, M., Pathirana, D., Chesham, J.E., Adamson, A., Spiller, D., Zindy, E., et al. (2020). Quantitative live imaging of Venus::BMAL1 in a mouse model reveals complex dynamics of the master circadian clock regulator. *PLoS Genet.* *16*, 1–24.

Reviewer comments, second round –

Reviewer #1 (Remarks to the Author):

This revised manuscript nicely addresses concerns from the first round of review with additional text, data analyses and modeling to bridge these new observations with prior data to set forth new questions that cut to the core of the mammalian circadian clock. Altogether, these data provide new evidence of, and new tools to explore, the delayed role of CRY1 in the clock. The questions raised here about the dependence of CRY1 on PER2 for nuclear entry and its activity outside of (and after) the large PER-CRY complexes that dominate the early part of repression should be of interest to the circadian community. I support publication in this journal.

Reviewer #2 (Remarks to the Author):

The authors have properly responded to all my previous comments and revised the manuscript accordingly. I have no further comments.

Reviewer #3 (Remarks to the Author):

The authors have adequately addressed all my concerns. They have performed extensive new analysis as I suggested, including modeling, FCS, evaluation of auto-fluorescence. I also agree with the authors that keeping both technical details and biological insights are indeed useful to the circadian clock field.

I recommend publication of the paper.

Reviewer #4 (Remarks to the Author):

This review was solicited to comment on how much the manuscript is improved by the added modeling (particularly in response to Reviewer 3). This is an interesting manuscript that could be use useful for modeling, I agree with Reviewer 3 that there is a missed opportunity. Timecourses contain much information about underlying systems and have played important roles in systems biology. Careful fitting to these has yielded very important information especially about cyclic systems such as cell cycle and circadian rhythms. New methods for understanding basic biology from careful analysis have been presented (Pigolotti et al. PNAS 2007). This work was inspired by timecourses such as presented for circadian rhythms by Lee et al. Cell 2001, and modeling of this data by Forger and Peskin PNAS 2003 predicted that the cryptochrome proteins could be in higher concentration in the nucleus than the period proteins despite the role of the period proteins shuttling them back and forth to the nucleus. The results of this manuscript confirm that prediction.

Reviewer 3 requests a careful analysis of this data with modeling but all that is shown is that a model can show that cryptochrome proteins can be at higher concentration than the period proteins. Little details of the model are given except a reference to a paper whose central point seems to be that the feedback loop studied in this manuscript is less important than originally thought (and that rhythms can be generated by a BMAL loop). That doesn't seem like what Reviewer 3 is asking for.

New modeling results would be most welcome. Making the data available and directly fitting them would also be helpful. Testing different hypothesis also might add strength. However, this would likely require a different approach than is presented here. Perhaps the authors could make their data publicly available and point to the many different types of analysis of these timeseries that

could be done?

Reviewer #4 (Remarks to the Author):

This review was solicited to comment on how much the manuscript is improved by the added modeling (particularly in response to Reviewer 3). This is an interesting manuscript that could be use useful for modeling, I agree with Reviewer 3 that there is a missed opportunity. Timecourses contain much information about underlying systems and have played important roles in systems biology. Careful fitting to these has yielded very important information especially about cyclic systems such as cell cycle and circadian rhythms. New methods for understanding basic biology from careful analysis have been presented (Pigolotti et al. PNAS 2007). This work was inspired by timecourses such as presented for circadian rhythms by Lee et al. Cell 2001, and modeling of this data by Forger and Peskin PNAS 2003 predicted that the cryptochrome proteins could be in higher concentration in the nucleus than the period proteins despite the role of the period proteins shuttling them back and forth to the nucleus. The results of this manuscript confirm that prediction.

Reviewer 3 requests a careful analysis of this data with modeling but all that is shown is that a model can show that cryptochrome proteins can be at higher concentration than the period proteins. Little details of the model are given except a reference to a paper whose central point seems to be that the feedback loop studied in this manuscript is less important than originally thought (and that rhythms can be generated by a BMAL loop). That doesn't seem like what Reviewer 3 is asking for.

New modeling results would be most welcome. Making the data available and directly fitting them would also be helpful. Testing different hypothesis also might add strength. However, this would likely require a different approach than is presented here. Perhaps the authors could make their data publicly available and point to the many different types of analysis of these timeseries that could be done?

We thank the Reviewer for her/his insightful comments and strongly agree that our experimental data and potential findings can benefit greatly from modeling efforts.

Here, we chose to conceptualize our main experimental findings by building-up on an existing model of the circadian oscillator. To fit the model to our new results, we modified it at three main stages:

- (i) We simplified the PER2-CRY1 loop by removing the PER2 phosphorylation module.
- (ii) We added a dissociation event of the nuclear PER2:CRY1 complex to the respective monomers.
- (iii) We did not assume degradation of the cytoplasmatic and nuclear PER2:CRY1 complexes, but rather assumed degradation of the monomers after dissociation of the complex.

We regret that in the previous version of our manuscript details of the model were not clearly presented. In the previous version, there was a typo in the main text that resulted in an incorrect reference to our Supplementary Note describing the model details (it should be Supplementary Note 2), for which we apologize.

Now, we try better describing the details in the main text by referring the readers to the following supplementary sections: We have presented the details of the resulting new model in **Supplementary Figure 9** and the corresponding modified equations in **Supplementary Note 2**. The names of the variables have been provided in **Supplementary Table 6**. We have described the model parameters, including the modified or newly introduced parameters, in **Supplementary Table 7**.

The Reviewer is also correct that we should have also better highlighted that this adapted model not only reproduces the higher CRY1 than PER2 total abundance levels, but also two additional important experimental findings: (i) the phase difference between PER2 and CRY1 and (ii) the much higher relative amplitudes of PER2 oscillations compared to CRY1 oscillations. The new model results (**Figure 5b**) reproduced all three experimental findings. Importantly, our new model makes additional predictions that can be experimentally tested: (i) The half-life of nuclear CRY1 is predicted to be higher than that of nuclear PER2. (ii) Association of cytoplasmatic PER2 and CRY1 should occur at a higher rate than the dissociation event and, conversely, in the nucleus, dissociation of the PER2:CRY1 complex should occur at a higher rate than association, driving the net PER2:CRY1 flux toward nuclear monomers. We have summarized these predictions in **Figure 5a**. Of note, in a number of studies estimating the half-lives of PER2 and CRY1, the values of the half-lives vary between studies, but the trend is consistent with our modeling predictions, *i.e.* that CRY1 is more stable than PER2 (*e.g.*, Smyllie et al., 2016 and Siepka et al., 2007).

In summary, our modified model reproduces three important experimental findings of our study (total abundance, phase and amplitude differences between PER2 and CRY1) and predicts additional differences in terms of half-life and binding properties that can be experimentally tested in future studies. **In the new version of the manuscript, we describe more explicitly that our motivation for modeling was driven by all of these findings simultaneously and not just the higher absolute CRY1 levels.**

In addition, we thank the Reviewer for her/his appreciation of our experimental time series data for use in mathematical modeling projects. We fully agree that we should make these data available to the scientific community for future computational studies. **We have therefore created a new Supplementary Table 10 containing all image-extracted time series data.**

The Reviewer's suggestion to compare our experimental data with mathematical models is also very stimulating. In the first version, we deliberately chose not to overload our manuscript with such analyses, because our manuscript already included two comprehensive parts: (i) the comprehensive description of an efficient strategy for generating knock-in reporter cell lines using CRISPR technology, which is particularly useful for genes expressed transiently or at low levels, such as those coding for circadian clock proteins; (ii) the use of this strategy to generate several knock-in cells and their careful characterization with various kinds of experimental techniques. At the request of Reviewer 3, we have added a small modeling section aimed at conceptualizing our three main experimental results (see above) in a way that is easily understood by the broader *Nature Communication* audience. Although this proposed direction is very exciting, an extensive data-driven modeling is beyond the scope of this manuscript.

Encouraged by this Reviewer's comments, we have performed several additional analyses and are now considering a follow-up project with extensive theoretical analysis. If the Reviewer and Editors deem it useful (and not too much) for this experiment-centered paper, we would like to offer to include the following results:

Since single cell data are notoriously noisy, direct fitting of gene-regulatory models to the raw time-series data is difficult. Therefore, we extracted a set of fundamental oscillator properties such as periods, amplitudes, and phase differences for modeling analysis. Given the enormous cell-to-cell variability in critical cellular processes, such as transcription (Raj et al., 2006), translation (Sonneveld et al., 2020), degradation (Aber et al., 2018), and nuclear translocation (Chang et al., 2017), we applied an ensemble approach: Using regression methods (MetaCycle; Wu et al., 2016) we extracted oscillatory parameters for all significantly rhythmic experimental time series and examined their interdependencies. We then compared these dependencies with model simulations. This approach

allowed us to test the following hypothesis: While for linear harmonic oscillators the periods do not depend on the amplitudes, we hypothesized that for circadian (non-linear) oscillators there are specific amplitude-period dependencies as described in the pioneering studies of Georg Duffing (1918). Indeed, analyzing the time series of 87 individual cells, we found a striking positive correlation of PER2 amplitude and period. Importantly, our modified model reproduced this dependence (we simulated cell-to-cell variability by randomly changing all transcription, translation, degradation, as well as translocation parameters by 10%). Such relationships between the amplitudes and periods of an oscillator are called "twist" and have been discussed previously (Winfree, 2001). Theory predicts that "hard oscillators" lead to negative correlations between periods and amplitudes, while "soft oscillators" exhibit positive correlations. Thus, our data suggest that the circadian oscillators studied belong to the class of soft oscillators.

We present these new data in the Results section (Figure 5c,d) and discuss them in the Discussion.

References:

- Alber AB, Paquet ER, Biserni M, Naef F, Suter DM. Single Live Cell Monitoring of Protein Turnover Reveals Intercellular Variability and Cell-Cycle Dependence of Degradation Rates. *Mol Cell* 71(6):1079-1091.e9 (2018).
- Chang AY, Marshall WF. Organelles - understanding noise and heterogeneity in cell biology at an intermediate scale. *J Cell Sci.* 130(5):819-826 (2017).
- Duffing G. *Erzwungene Schwingungen bei veränderlicher Eigenfrequenz und ihre technische Bedeutung.* Vieweg Braunschweig (1918).
- Raj A, Peskin CS, Tranchina D, Vargas DY, Tyagi S. Stochastic mRNA synthesis in mammalian cells. *PLoS biology* 4(10):e309 (2006)
- Siepkha, S. M. et al. Circadian Mutant Overtime Reveals F-box Protein FBXL3 Regulation of Cryptochrome and Period Gene Expression. *Cell* 129, 1011–1023 (2007).
- Smyllie, N. J. et al. Visualizing and Quantifying Intracellular Behavior and Abundance of the Core Circadian Clock Protein PERIOD2. *Curr. Biol.* 26, 1880–1886 (2016).
- Sonneveld S, Verhagen BMP, Tanenbaum ME. Heterogeneity in mRNA Translation. *Trends Cell Biol.* 30(8):606-618 (2020).
- Winfree, A. T. *The geometry of biological time: Interdisciplinary applied mathematics.* (Springer New York, 2001).
- Wu, G., Anafi, R. C., Hughes, M. E., Kornacker, K. & Hogenesch, J. B. MetaCycle: an integrated R package to evaluate periodicity in large scale data. *Bioinformatics* 32, 3351–3353 (2016).

Reviewer comments, third round –

Reviewer #4 (Remarks to the Author):

I would like to thank the authors for clarifying their modeling work and providing new data showing how the amplitude period of the clock increases with increasing period. I wholeheartedly encourage the publication of this manuscript in Nature Communications, and believe, as I did before, that this work is very important for the field.